# Autoregressive Boltzmann Generators

**Danyal Rehman** [1 2 3 4]   **Charlie B. Tan** [1 5]   **Yoshua Bengio** [1 4 6]   **Avishek Joey Bose** [1 7 *]   **Alexander Tong** [3 *]

## Abstract

Efficient sampling of molecular systems at thermodynamic equilibrium is a hallmark challenge in statistical physics. This challenge has driven the development of Boltzmann Generators (BGs), which allow rapid generation of uncorrelated equilibrium samples by combining a generative model with exact likelihoods and an importance sampling correction. However, modern BGs predominantly rely on normalizing flows (NFs), which either suffer from limited expressivity due to strict invertibility constraints (discrete time) or computationally expensive likelihoods (continuous time). In this paper, we propose AUTOREGRESSIVE BOLTZMANN GENERATORS (ARBG)—a novel autoregressive modelling framework—that overcomes these limitations by departing from the flow-based BG paradigm. ARBG circumvents the topological constraints of flows and enables sequential inference-time interventions, while offering enhanced scalability by leveraging architectures effective in Large Language Models. We empirically demonstrate that ARBG leads to significant improvements over flow-based models across all benchmarks, but particularly in larger peptide systems such as the 10-residue Chignolin. Furthermore, we introduce ROBIN, a 132 million parameter transferable model trained with the ARBG framework which improves over the previous state-of-the-art, reducing the zero-shot energy error, $\mathcal{E}$-$\mathcal{W}_2$, on 8-residue systems by over 60%.

## 1. Introduction

A central insight of statistical mechanics is that macroscopic phenomena—such as protein folding (Noé et al., 2009;

[1] Mila – Québec AI Institute [2] Broad Institute of MIT & Harvard [3] Aithyra [4] Université de Montréal [5] University of Oxford [6] CIFAR Senior Fellow [7] Imperial College London. Correspondence to: Danyal Rehman <danyal.rehman@mila.quebec>, Alexander Tong <atong@aithyra.at>.

*Proceedings of the 43rd International Conference on Machine Learning*, Seoul, South Korea. PMLR 306, 2026. Copyright 2026 by the author(s).

Lindorff-Larsen et al., 2011), magnetization of an Ising model (Yang, 1952), and crystal structure formation (Parrinello & Rahman, 1980; Matsumoto et al., 2002)—are governed by the ensemble of microscopic states at equilibrium. This equilibrium distribution is known as the *Boltzmann distribution*: $\mu_{\text{target}}(x) \propto \exp\left(-\mathcal{E}(x)/k_B T\right)$, where $\mathcal{E}(x)$ is the dimensionless potential energy of a conformation $x \in \mathbb{R}^{n \times 3}$, and $k_B T$ is the thermal energy at absolute temperature $T$. Accordingly, the computational challenge is to efficiently draw statistically-independent samples from this target distribution.

A key characteristic of this sampling problem is that states in thermodynamic equilibrium—i.e., modes of the distribution—are often sparse and well-separated by high-energy barriers (Wirnsberger et al., 2020; Rizzi et al., 2021). The dominant approach for exploring this conformational landscape is through Molecular Dynamics (MD) simulations (Alder & Wainwright, 1959; Rahman, 1964), which seek to simulate the equations of motion with finely-discretized time-steps. However, this approach suffers from a severe timescale issue. Specifically, MD simulations often use time-steps on the order of femtoseconds ($10^{-15}$s), but mode mixing across these high-energy barriers typically requires time-resolutions of microseconds ($10^{-6}$s) to seconds ($10^{0}$s) (Olsson, 2026). Consequently, the vast majority of MD computation is spent simulating high-frequency vibrations within local minima rather than exploring the global energy landscape, rendering MD too computationally expensive for practical problems (Perez et al., 2025). While many accelerated MD schemes have been explored (Hénin et al., 2022; Syed et al., 2021; Klein et al., 2023a; Kapuśniak et al., 2026), they still have difficulty with this fundamental mixing problem.

Boltzmann Generators (BGs) (Noé et al., 2019) have emerged as a powerful framework to circumvent this. BGs learn a generative model $p_\theta(x)$ to propose samples for importance sampling, leveraging the exact model likelihood $p_\theta(x)$ and the target energy $\mathcal{E}(x)$. This allows for the parallel generation of independent and consistent samples without having to traverse between modes. These features make BGs an attractive framework when performing equilibrium sampling of large molecular systems. However, to satisfy the requirement for tractable exact likelihoods, the field has relied almost exclusively on normalizing

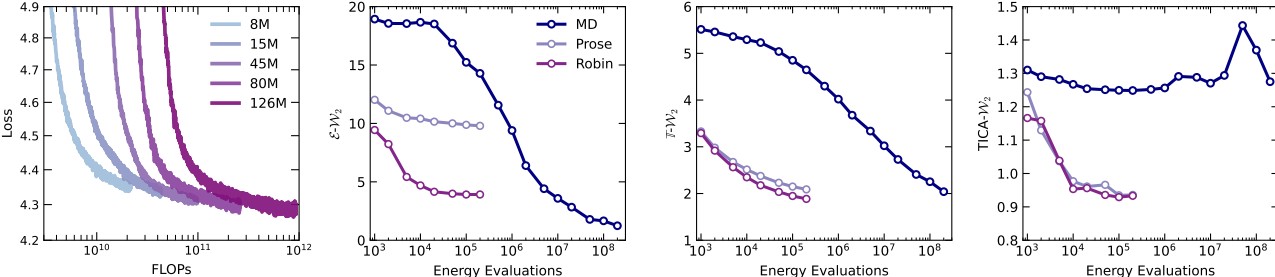

*Figure 1.* **(Left)** The training loss curves of models of varying scale (ranging from 8M to 126M parameters) as a function of the total number of floating point operations (FLOPs) on the decapeptide: Chignolin. **(Right)** Global and local performance metrics as a function of the number of energy evaluations demonstrating inference-time scaling between our transferable model ROBIN, the previous state-of-the-art BG Prose, and a short molecular dynamics (MD) chain for the same number of energy function evaluations.

flows (NFs) in either discrete (Tabak & Vanden-Eijnden, 2010; Tabak & Turner, 2013; Dinh et al., 2017; Rezende & Mohamed, 2015; Tan et al., 2025a; Rehman et al., 2026b) or continuous time (Chen et al., 2018; Rehman et al., 2026a).

This reliance on flow-based architectures, however, imposes severe theoretical limitations. Fundamentally, NFs in practical instantiations are not only diffeomorphisms but also *homeomorphisms*, as the prior is a single Gaussian (Cornish et al., 2020; Dupont et al., 2019; Runde et al., 2005). Consequently, such generative models preserve the topology of their domain and struggle to model target distributions with distinct topologies to the prior, e.g., disjoint supports, differing number of connected components, or "holes". The equilibrium distribution of molecular conformations precisely exhibits these challenging topologies, as several metastable states are separated by regions of high-energy barriers. To morph the single connected mode of a Gaussian to these effectively separated states, a flow is forced to perform extreme deformations, stretching space across thin bridges and compressing it into modes. This leads to highly non-smooth mappings prone to exploding Lipschitz constants and ill-conditioned Jacobians, resulting in discrete time NFs being notoriously unstable, greatly limiting model expressivity.

Continuous normalizing flows (CNFs) (Chen et al., 2018) are free from the same architectural constraints as discrete-time NFs. In addition, modern training strategies for flow-matching (Peluchetti, 2021; Liu, 2022; Lipman et al., 2023; Albergo & Vanden-Eijnden, 2023) can greatly stabilize the optimization process, yet the ill-conditioning manifests itself during inference. The learned vector field becomes highly non-smooth, resulting in *stiff* dynamics (Hochbruck & Ostermann, 2010; Hochbruck et al., 2020) leading to a large number of function evaluations being required at inference for accurate likelihoods to be obtained. This presents a currently unavoidable tradeoff: (**1**) Discrete-time NFs yield efficient likelihoods but struggle with poor sample quality and limited expressivity, while (**2**) CNFs are more expressive, but require expensive ODE solvers

for accurate likelihood evaluation, rendering importance sampling approaches computationally prohibitive.

**Present work**. In this work, we propose AUTOREGRESSIVE BOLTZMANN GENERATORS (ARBG), a novel alternative to flow-based BGs that circumvents these limitations. ARBG employs an autoregressive paradigm to factorize the molecular density into a sequence of conditional densities: $p_\theta(x) = \prod_j p(x_j|x_{<j})$. This formulation offers three distinct advantages: (**1**) ARBG overcomes the expressivity bottlenecks of discrete time flows without incurring the computational cost of continuous time flows; (**2**) ARBG avoids the numerical instability of learning high-distortion diffeomorphisms, allowing it to model discontinuous jumps and separate modes present in complex multi-modal target densities; and (**3**) ARBG benefits from the investment and advances in discrete generative modelling that power modern LLMs, and exhibits similar scaling properties in both model size and inference samples (Figure 1).

Our main contributions are summarized as follows:

- We introduce AUTOREGRESSIVE BOLTZMANN GENERATORS (ARBG), the first scalable, autoregressive and diffeomorphism-free method for Boltzmann Generation.

- We investigate various proposal formulations, demonstrating that discrete binning not only offers superior training stability and scalability compared to continuous mixture models, but also unlocks sequential inference-time interventions that are not possible in flow-based architectures.

- We demonstrate that ARBG consistently outperforms all baseline methods across every single-peptide benchmark, with especially strong performance and scalability demonstrated on the 10-residue Chignolin system (Figure 1).

- We introduce ROBIN, a 132 million parameter transferable autoregressive generative model that achieves zero-shot generalization to unseen peptides, reducing Energy-W2 error, $\mathcal{E}\text{-}\mathcal{W}_2$, by over $60\%$ compared to the previous state-of-the-art approach on large peptides: Prose, a discrete-time NF (Tan et al., 2025b).

## 2. Background and Preliminaries

**Thermodynamic Equilibrium Sampling**. We consider molecular systems comprising of $n$ atoms, represented at all-atom resolution by its conformations $x \in \mathbb{R}^{n \times 3}$. The equilibrium behaviour of the system is characterized by the following target Boltzmann distribution:

$$\mu_{\text{target}}(x) \propto \exp\left(\frac{-\mathcal{E}(x)}{k_{\text{B}}T}\right), \ \mathcal{Z} = \int_{\mathcal{X}} \exp\left(\frac{-\mathcal{E}(x)}{k_{\text{B}}T}\right) dx.$$

Here, $\mathcal{E} : \mathbb{R}^{n \times 3} \to \mathbb{R}$ denotes the potential energy, with the exponent containing the thermal factor $k_{\text{B}}T$. We can absorb $k_{\text{B}}T$ into $\mathcal{E}$. The distribution is normalized by the partition function $\mathcal{Z}$, which is computationally intractable to evaluate exactly. Macroscopic properties are obtained by computing observables $\phi(x)$ as expectations with respect to the Boltzmann distribution $\mu_{\text{target}}(x)$. A commonly-used approach to obtain consistent samples from the Boltzmann distribution leverages self-normalized importance sampling (SNIS) with an easy-to-sample proposal distribution $p(x)$:

$$\mathbb{E}_{\mu_{\text{target}}(x)}\left[\phi(x)\right] = \mathbb{E}_{p(x)}\left[\phi(x)\bar{w}(x)\right] \approx \frac{\sum_{i=1}^{K} w(x^i)\phi(x^i)}{\sum_{i=1}^{K} w(x^i)},$$

where $w(x^i) = \exp\left(-\mathcal{E}(x^i)\right)/p(x^i)$ is the unnormalized importance weight and $x^i \sim p(x), i \in [K]$ are $K$ statistically-independent samples from the proposal $p(x)$. It is well known that SNIS is a consistent estimator whose variance depends on the distributional overlap of the chosen proposal $p(x)$ compared to $\mu_{\text{target}}(x)$ (Owen, 2013).

**Boltzmann Generators**. A Boltzmann Generator (Noé et al., 2019) learns a parameterized proposal distribution $p_\theta(x)$ that serves to approximate the target distribution $\mu_{\text{target}}(x)$. Crucially, $p_\theta(x)$ admits a tractable likelihood for samples $x \sim p_\theta(x)$, enabling the computation of importance weights required for self-normalized importance sampling. For instance, when $p_\theta(x)$ is a normalizing flow defined by a composition of invertible maps $f_\theta = f_M \circ \cdots \circ f_1$, the likelihood can be computed by the change-of-variables formula $\log p_\theta(x_M) = \log p_0(x_0) - \sum_{i=1}^{M} \log |\partial f_{i,\theta}(x_{i-1})/\partial x_{i-1}|$, with $x_i = f_i(x_{i-1})$, and crucially $f_i$ being invertible.

In cases where obtaining a likelihood is theoretically possible, but computationally expensive—such as in CNFs—an SNIS-based resampling step becomes equally impractical. To see this more clearly, we can detail the Augmented ODE of $3n + 1$ dimensions, which tracks both the particle evolution $x_t$ across simulation time $t \in [0, 1]$ using the learned velocity field associated with the CNF $v_\theta(x_t, t)$, and the corresponding evolution of the induced log-density $\log p_{t,\theta}(x_t)$. We can simulate this trajectory from time $t = 0$ to time $t = 1$ by sampling from a prior distribution $x_0 \sim p_0(x_0)$ and then integrating along the Augmented ODE:

$$\begin{bmatrix} x_1 \\ \log p_{1,\theta}(x_1) \end{bmatrix} = \begin{bmatrix} x_0 \\ \log p_0(x_0) \end{bmatrix} + \int_0^1 \begin{bmatrix} v_\theta(x_t, t) \\ -\nabla \cdot v_\theta(x_t, t) \end{bmatrix} dt.$$

Here $\nabla\cdot$ is the divergence operator, which requires $O(d)$, with $d = n \times 3$, function evaluations of $v_\theta$ per-integration step. While faster unbiased estimators of the divergence, such as the Hutchinson trace estimator (Meyer et al., 2021) exist, the added variance incurred renders them unsuitable for the Boltzmann Generator setting (Klein et al., 2023b).

## 3. AUTOREGRESSIVE BOLTZMANN GENERATORS

We now consider an alternative class of generative models for constructing the proposal distribution $p_\theta(x)$ in a Boltzmann Generator—autoregressive (AR) models. We introduce AUTOREGRESSIVE BOLTZMANN GENERATORS, the first diffeomorphism-free AR model for molecular systems that operates directly on atoms in their native Cartesian coordinates. In the context of Boltzmann Generators, the central challenge of AR modelling arises from the continuous nature of molecular configurations, in contrast to the discrete data on which AR models are most frequently employed. This setting, therefore, presents an opportunity for developing novel AR frameworks for continuous-state systems. We further motivate our model choice by identifying two key properties required of a proposal distribution within a Boltzmann Generator:

**(1) Fast and Accurate Likelihood**. A crucial requirement of any proposal in a BG is to facilitate SNIS-based correction of samples—necessitating access to fast, unbiased, and preferably exact likelihood evaluation $p_\theta(x)$.

**(2) Scalability**. We require expressive generative model families $p_\theta(x)$ that can capture the complex, sparse, and rugged energy landscape of high-dimensional molecular systems with predictable scaling behaviour.

We highlight that both desiderata are demonstrably satisfied by autoregressive models, but not necessarily flow-based models. In particular, autoregressive models *exactly* factorize the joint density over $x$ as a sequence of conditional distributions that is used to predict the next dimension conditioned on the history $x_{<j} = [x_1, \ldots x_{j-1}]$, for $j \in [d]$:

$$\log p_\theta(x) = \sum_{j=1}^{d} \log p_\theta(x_j | x_{<j}). \tag{1}$$

Clearly, Eq. 1 allows for exact likelihood computation in a single pass, avoiding Jacobian determinants or computationally expensive numerical solvers for Neural ODEs. Moreover, AR models have achieved empirical success across a spectrum of domains at scale, including large-scale discrete modelling of text (Comanici et al., 2025) and images (Dosovitskiy et al., 2021). Indeed, the diversity of data domains tackled by autoregressive models comes without specific constraints, such as invertible architectures, or the need to model diffeomorphisms. The latter fact we argue is particularly important for molecular modelling due

to the non-smooth nature of the target Boltzmann $\mu_{\text{target}}(x)$.

We next outline how to build ARBG in §3.1, and explore new unlocked capabilities of an AR model for BGs in §3.2.

### 3.1. Autoregressive Modelling of Conformations

We consider inputs of the form $x \in \mathbb{R}^{n \times 3}$, which are flattened into a single $d = n \times 3$ dimensional vector by an autoregressive model as defined in Equation (1). From here onwards, we use subscripts such as $x_j$ to denote the $j$-th dimension of the input vector $x$ rather than a continuous time index as done for CNFs. As AR models require an ordering to model molecular states, we use a residue-by-residue ordering in which the sidechains for each residue immediately follow the backbone atoms.

A key technical challenge of instantiating AR models over continuous spaces is determining the parametrization of the conditional distribution over dimensions $p_\theta(x_j|x_{<j})$. We explore several options by leveraging existing ideas from Mixture Density Networks (MDN) (Bishop, 1994), which output the parameters of the conditional distribution as a mixture. In addition, we also offer a novel parametrization utilizing a uniform binning strategy that is simple yet enjoys closer alignment to LLM training—unveiling predictable scaling behaviours but now in the context of BGs.

**Conditionals as Discretized Mixtures**. Following the pioneering work of PixelCNN++ (Salimans et al., 2017), we demonstrate how to adapt such an approach to model molecular conformations, leading to MoL-PixelCNN++ and a novel extension in GMM-PixelCNN++. These serve as modernized instantiations of the MDN for molecular conformations and later as baselines in our experiments §4.

To construct these conditional mixtures, we model the input $x_j$, e.g. a *singular* spatial dim of an atom, by first finely discretizing the space into $B$ uniform bins of width $\Delta$ over a pre-determined interval range $\mathcal{I} = [C_{\min}, C_{\max})$, with range cutoffs $C_{\min}$ and $C_{\max}$ picked through a data standardization step. Formally, the mapping $\mathcal{Q} : \mathbb{R} \to [L]$ from a continuous coordinate $x_j \in \mathbb{R}$ to a bin index $b_l \in [L]$ is given by:

$$b = \mathcal{Q}(x_j) = \left\lfloor \frac{\tilde{x}_j - C_{\min}}{\Delta} \right\rfloor, \ \tilde{x}_j = \texttt{clip}(x_j, C_{\min}, C_{\max}).$$

After binning, we assume a latent assignment $k$ sampled from a $K$-component mixture, $\pi_k \sim \text{Cat}(\pi_1, \ldots, \pi_K)$. Each mixture component can then be prescribed by an easy-to-parametrize distribution, such as the logistic distribution, leading to a Mixture of Logistics (MoL) or a Gaussian Mixture Model (GMM). We can compute the discretized probability mass that an observed value $x_j$ falls in bin $b$, i.e. $p(X = x_j|x_{<j})$, by leveraging the integral of the CDF differences of the logistic distribution at $\tilde{x}_j + \Delta/2$ and $\tilde{x}_j - \Delta/2$. This leads to the following

parameterizations of the conditional $p_\theta(x_j|x_{<j})$:

$$\sum_{k=1}^{K} \pi_k \underbrace{\left[ \varphi\left( \frac{\tilde{x}_j + \Delta/2 - \mu_k}{\sigma_k} \right) - \varphi\left( \frac{\tilde{x}_j - \Delta/2 - \mu_k}{\sigma_k} \right) \right]}_{p_k(\tilde{x}_j \in b)}$$

$$\text{where } \pi_k \geq 0, \sum_{k=1}^{K} \pi_k = 1, \sigma_k > 0. \qquad (2)$$

$\pi_k$ is the weight of the $k$-th mixture component, $\varphi$ is the logistic function or the standard normal CDF, i.e., $\varphi(z) = 0.5\left(1 + \text{erf}\left(z/\sqrt{2}\right)\right)$, for the MoL and GMM cases, respectively. The edge cases of the first and last bin for the MoL-PixelCNN++ are handled by replacing $x_j - \Delta/2$ and $x_j + \Delta/2$ by $-\infty$ and $+\infty$, respectively. Finally, we can also handle the edge case for GMM-PixelCNN++'s leftmost and rightmost bins:

$$p_k(\tilde{x}_j \in b_0) = \Phi\left( \frac{\tilde{x}_j + \Delta/2 - \mu_k}{\sigma_k} \right),$$

$$p_k(\tilde{x}_j \in b_L) = 1 - \Phi\left( \frac{\tilde{x}_j - \Delta/2 - \mu_k}{\sigma_k} \right).$$

In both cases, the conditional mixtures admit an analytic log-likelihood, enabling conventional MLE-based training.

**Uniform Bin Parameterization**. While elegant in theory, Mixture Density Networks like the MoL-PixelCNN++ and GMM-PixelCNN++ are prone to mode collapsing of the mixture components $\pi_k$ to a small subset (Deng et al., 2022), potentially leading to suboptimal performance on harder systems of interest. We remedy this problem by abandoning mixture models altogether and introducing, arguably, the simplest parameterization of $p_\theta(x_j|x_{<j})$ by predicting directly the bin centres $b_l$ as a Categorical distribution during training. This allows us to directly use an autoregressive model, as commonly done for LLMs for next prediction, but now for molecular data. At inference, to recover a continuous coordinate, we can simply add uniform noise to the sampled bin centre: $x_j = b_l + u_l$, where $u_l \sim \text{Unif}(-\Delta/2, \Delta)$. Such a uniform bin parameterization induces the following piecewise-conditional density:

$$p_\theta(\tilde{x}_j|x_{<j}) = \sum_{l=1}^{L} \pi_\theta(b_l|x_{<j}) \frac{\mathbf{1}\{\tilde{x}_j \in b_l\}}{\Delta}, \qquad (3)$$

where $\pi_\theta(b_l|x_{<j}) = \text{Cat}(b_0, \ldots, b_L)$. In Equation (3) we observe that when $\Delta$ is the same for all bins—i.e., uniform binning—then the conditional log density has a constant offset $\log \Delta$ which vanishes with the number of bins.

Under the uniform bin parameterization, we are also able to quantify the exact log-likelihood error. Let $p^\star(\tilde{x}_j|x_{<j})$ denote the true conditional density and define its bin masses

$$p^\star(b_l|x_{<j}) := \int_{b_l} p^\star(u|x_{<j}) \, du.$$

Also, define the true conditional density restricted to a bin:

$$p^\star(\tilde{x}_j | \tilde{x}_j \in b_l, x_{<j}) := \frac{p^\star(\tilde{x}_j | x_{<j})}{p^\star(b_l | x_{<j})} \mathbf{1}\{\tilde{x}_j \in b_l\}.$$

The lowest attainable error of $p_\theta(x_j | x_{<j})$ as measured by the KL-divergence, is given by the following proposition.

**Proposition 1.** *Let the true conditional density be given by $p^*(x_j | x_{<j})$ and the autoregressive model's conditional density $p_\theta(x_j | x_{<j})$ under the uniform bin parameterization. The resulting minimum achievable error of the autoregressive model in KL is,*

$$\inf_\theta \mathbb{D}_{\mathrm{KL}}\left(p^*(x_j | x_{<j}) \| p_\theta(x_j | x_{<j})\right) =$$

$$\sum_{l=1}^{L} p^\star(b_l | x_{<j}) \mathbb{D}_{\mathrm{KL}}\left(p^\star(\tilde{x}_j | \tilde{x}_j \in b_l, x_{<j}) \| \mathrm{Unif}(\Delta)\right).$$

Proposition 1 quantifies the intrinsic loss of modelling resolution induced by piecewise-uniform de-quantization within bins. Intuitively, it explicates that the mismatch corresponds to precisely the true density within each bin not being exactly uniform with width $\Delta$, which is precisely the irreducible error of this AR model parameterization. Moreover, we highlight the KL term is merely the *Shannon entropy* of $p^\star(\tilde{x}_j | \tilde{x}_j \in b_l, x_{<j})$ up to a constant as we are comparing it against the uniform distribution $\mathrm{Unif}(\Delta)$. In Figure A.1 and Figure A.2 in the Appendix, we quantify the impact of uniform binning for single peptide systems we consider by analyzing the distribution of coordinates in each bin index as well as irreducible error due to this distributional mismatch. Importantly, choosing the appropriate number of bins *still allows us* to perform effective resampling through SNIS as we demonstrate in our experiments §4.

### 3.2. Tools Enabled by ARBG

ARBG's autoregressive factorization unlocks a large toolkit of inference-time interventions unavailable to flow-based approaches that generate all data dimensions concurrently. By decomposing generation into discrete steps over each conditional, we can leverage standard techniques from modern LLMs, such as temperature scaling for diversity control, and extend them to the molecular domain. Crucially, this autoregressive structure admits intermediate intervention; we can interrogate, correct, or discard partial conformations before the full molecule is realized. We proceed to demonstrate this capability via a granular resampling scheme.

**Autoregressive Twisted Sequential Monte Carlo**. SMC sampling has been a staple tool across varied domains (Doucet et al., 2001; Del Moral et al., 2006). In ARBG, we can improve efficiency by early exiting the sampling process on physically implausible substructures (e.g., steric clashes). We implement this via an Autoregressive Twisted SMC. We define a series of *twist* functions

---

**Algorithm 1** Autoregressive SMC Inference

---
**Require:** Pre-trained proposal $p_\theta$, batch size $M$, and energy for $s$-length capped residues $\mathcal{E}_s$
1: $w_0 \leftarrow 0$
2: **for** $j$ in $1, \ldots, d$ **do**
3:      **for** $m$ in $1, \ldots, M$ **do**
4:          $x_j^m \sim p_\theta(x_j^m | x_{<j}^m)$
5:          $w_j^m \leftarrow w_{j-1}^m \left( \frac{\psi_j(x_j^m)}{\psi_{j-1}(x_{j-1}^m)} \right)$
6:      **end for**
7:      **if** IS_RESIDUE_END$(j)$ **then**
8:          $x_{\leq j} \leftarrow$ SYSTEMATIC-RESAMPLE$(x_{\leq j}, w_j)$
9:          $w_j \leftarrow 0$
10:     **end if**
11: **end for**

---

$\psi_j(x_j)$ which define a number of intermediate densities $\eta_j(x_j) := p_\theta(x_j | x_{<j}) \psi_j(x_j)$. Instead of sampling directly from $p_\theta(x_j | x_{<j})$ at every step, the goal is to sample from the twisted intermediate distribution $\eta_j$. This can be accomplished by first calculating the likelihoods and the twist function values, then performing resampling (Algorithm 1). In our molecular case, we define the twist functions using partial energy evaluations:

$$\psi_j(x_j) = \frac{\exp(-\mathcal{E}_s(r_s(x_{\leq j})))}{p_\theta(r_s(x_{\leq j}) | r_{s-1}(x_{\leq j}))}, \tag{4}$$

where $r(x_{\leq j})$ defines the largest (capped) residue subset of $x_{\leq j}$, inclusive of $x_j$ (c.f. §E.5.1 for details on capped residues). This twist function allows us to inject physical validity constraints using any molecular energy function $\mathcal{E}_s$, a version of the original $\mathcal{E}$, that operates on peptides of length $s < d$. Whilst resampling can be done at any point, we tailor our SMC to the molecular setting by invoking systematic resampling (Douc & Cappé, 2005), at the end of the atomistic coordinates of a generated residue. We utilize residue-level granularity to leverage standard peptide force-fields, though finer atom-level twists are also possible and allow for earlier rejection. This approach is fundamentally distinct from prior flow-based resampling methods like SBG (Tan et al., 2025a). While SBG must generate complete candidates before resampling, meanwhile, ARBG enables substructure-level steering, correcting the generative process as soon as an error is detected.

## 4. Experiments

In this section, we empirically validate the efficacy of AUTOREGRESSIVE BOLTZMANN GENERATORS across a variety of molecular conformation sampling tasks. Our evaluation focuses on assessing the framework's scalability on single peptides systems ranging from alanine dipeptide to the 10-residue Chignolin in the same experimental data configuration of Tan et al. (2025a). We also eval-

*Table 1.* Results on Tri-alanine (AL3), Alanine tetrapeptide (AL4), Hexa-alanine (AL6), and Chignolin (GYDPETGTWG). Evaluations is performed with $2 \times 10^5$ energy evaluations; all methods except SBG use SNIS. Best values in **bold**, with second-best underlined.

| Algorithm | Tri-alanine (AL3) | | Tetrapeptide (AL4) | | Hexa-alanine (AL6) | | Chignolin (GYDPETGTWG) | |
| --- | --- | --- | --- | --- | --- | --- | --- | --- |
| | $\mathcal{E}\text{-}\mathcal{W}_2 \downarrow$ | $\mathbb{T}\text{-}\mathcal{W}_2 \downarrow$ | $\mathcal{E}\text{-}\mathcal{W}_2 \downarrow$ | $\mathbb{T}\text{-}\mathcal{W}_2 \downarrow$ | $\mathcal{E}\text{-}\mathcal{W}_2 \downarrow$ | $\mathbb{T}\text{-}\mathcal{W}_2 \downarrow$ | $\mathcal{E}\text{-}\mathcal{W}_2 \downarrow$ | $\mathbb{T}\text{-}\mathcal{W}_2 \downarrow$ |
| ECNF++ | $2.206 \pm 0.813$ | $0.962 \pm 0.253$ | $5.638 \pm 0.483$ | $1.002 \pm 0.061$ | $10.668 \pm 0.285$ | $1.902 \pm 0.055$ | — | — |
| RegFlow | $0.853 \pm 0.105$ | $1.577 \pm 0.140$ | $3.277 \pm 0.546$ | $2.342 \pm 0.102$ | — | — | — | — |
| SBG | $0.598 \pm 0.084$ | $0.503 \pm 0.029$ | $1.007 \pm 0.382$ | $1.039 \pm 0.069$ | $1.189 \pm 0.357$ | $1.444 \pm 0.140$ | $10.819 \pm 7.206$ | $3.778 \pm 0.440$ |
| FALCON-A | $1.385 \pm 0.182$ | $0.343 \pm 0.004$ | $2.929 \pm 0.068$ | $1.094 \pm 0.034$ | $1.211 \pm 0.105$ | $1.163 \pm 0.112$ | — | — |
| FALCON | $0.544 \pm 0.013$ | $0.452 \pm 0.011$ | $0.686 \pm 0.047$ | $0.858 \pm 0.077$ | $0.892 \pm 0.311$ | $1.256 \pm 0.132$ | — | — |
| GIVT | $1.354 \pm 0.058$ | $0.343 \pm 0.008$ | $1.033 \pm 0.449$ | $1.113 \pm 0.100$ | $1.206 \pm 0.056$ | $1.527 \pm 0.048$ | $45.646 \pm 20.989$ | $3.031 \pm 0.098$ |
| MoL-PixelCNN++ | $0.506 \pm 0.082$ | $1.024 \pm 0.686$ | $1.643 \pm 0.504$ | $1.415 \pm 0.110$ | $1.429 \pm 0.186$ | $1.264 \pm 0.205$ | $140.717 \pm 49.113$ | $3.391 \pm 0.093$ |
| GMM-PixelCNN++ | $0.249 \pm 0.025$ | $0.364 \pm 0.016$ | $1.434 \pm 0.783$ | $0.806 \pm 0.056$ | $1.164 \pm 0.037$ | $1.285 \pm 0.058$ | $23.339 \pm 6.485$ | $3.007 \pm 0.086$ |
| ARBG (ours) | $\mathbf{0.202 \pm 0.010}$ | $\mathbf{0.312 \pm 0.003}$ | $\mathbf{0.449 \pm 0.030}$ | $\mathbf{0.592 \pm 0.010}$ | $\mathbf{0.328 \pm 0.122}$ | $\mathbf{1.094 \pm 0.052}$ | $\mathbf{1.723 \pm 0.075}$ | $\mathbf{2.632 \pm 0.044}$ |

uate the zero-shot generalization capabilities on unseen sequences using our transferable model, ROBIN, on the ManyPeptidesMD introduced in (Tan et al., 2025b).

**Baselines**. We consider a suite of prior baselines that include the equivariant CNF (Klein et al., 2023b; Klein & Noe, 2024), using the ECNF++ results of Tan et al. (2025a). We additionally compare against discrete normalizing flows, in RegFlow (Rehman et al., 2026b), and the prior state-of-the-art method for single-system Boltzmann sampling SBG (Tan et al., 2025a). We also benchmark performance relative to few-step CNFs, i.e., flow-maps (Boffi et al., 2025; Geng et al., 2025): FALCON/FALCON-A, which differ in their training objectives (Rehman et al., 2026a). For ALDP, we further include BoltzNCE, an energy-based model trained via noise-contrastive estimation (Aggarwal et al., 2025). We further train a GIVT (Tschannen et al., 2024), and MoL-PixelCNN++ and GMM-PixelCNN++ as described in Section 3.1, following the same training procedure as ARBG. For the transferable setting, we compare against Timewarp (Klein et al., 2023a), BioEmu (Lewis et al., 2025), UniSim (Yu et al., 2025), TarFlow (Zhai et al., 2025), and the prior SOTA for transferable Boltzmann generation in Prose (Tan et al., 2025b).

**Metrics**. We evaluate our models with three complementary Wasserstein-based metrics following prior work. (1) The 2-Wasserstein energy distance ($\mathcal{E}\text{-}\mathcal{W}_2$), which measures agreement between generated and reference energy distributions, providing a sensitive test of local physical accuracy and consistency with the target Boltzmann distribution. (2) To assess structural coverage, we compute a 2-Wasserstein distance in torsional space ($\mathbb{T}\text{-}\mathcal{W}_2$) that respects angular periodicity, capturing global conformational differences and missing modes that may not be reflected in energies alone. While this torus-based metric is effective at detecting large-scale structural mismatches, it can be insensitive to rare mode loss. (3) Lastly, we analyze a time-lagged independent component analysis (TICA)-based 2-Wasserstein distance (TICA-$\mathcal{W}_2$) that compares samples in a lower-dimensional space spanned by the slowest dynamical modes fit on a reference trajectory. We exclude Effective Sample Size (ESS) (Kish, 1957) as a primary

metric, as it is incompatible with SMC-based schemes while also disproportionately rewarding "mode-seeking" models that collapse into single energy minima to minimize variance (Blessing et al., 2024). We instead prioritize metrics that penalize mode-dropping to ensure accurate global distributional coverage (see §C.2 for a detailed discussion).

### 4.1. Single Peptide Systems

We evaluate the performance of ARBG on conformation sampling tasks for single peptides, ranging from the simple alanine dipeptide (ALDP) (2 residues) to the large Chignolin (10 residues). We report our results in Table 1, and defer the ALDP results to Table A.3 in §D.3 due to both the simplicity of the dataset and also problems with the dataset construction leading to mode-collapse of models. We find that ARBG comprehensively outperforms on both $\mathcal{E}\text{-}\mathcal{W}_2$ and $\mathbb{T}\text{-}\mathcal{W}_2$ for all considered systems. We further observe MDN baselines like GMM-PixelCNN++ and GIVT achieve second place, suggesting the overall power of AR models for BG's in comparison to flow-based BGs.

**Scaling to Decapeptides**. SBG (Tan et al., 2025a) first demonstrated that discrete NFs can be scaled to the decapeptide Chignolin—a particularly challenging molecular structure due to the existence of the $\beta$-hairpin secondary structure. As shown in Table 1, ARBG significantly outperforms the SBG approach across all global and local evaluation metrics, further demonstrating its scalability. Additionally, Figure 2 shows that the reweighted energy distribution proposed by our model closely matches that of MD simulation data. Finally, the Ramachandran plots (Ramachandran et al., 1963) in Figure 3 indicate that our model accurately captures nearly all the conformational modes present in the test set.

### 4.2. Transferable Generation

We now introduce ROBIN, a transferable model trained using the ARBG framework with additional conditioning information—detailed in §E.2—to allow zero-shot transfer to unseen peptides in the ManyPeptidesMD dataset (Tan et al., 2025b). We report our results in Table 2, which contain test performance metrics averaged over 30 different

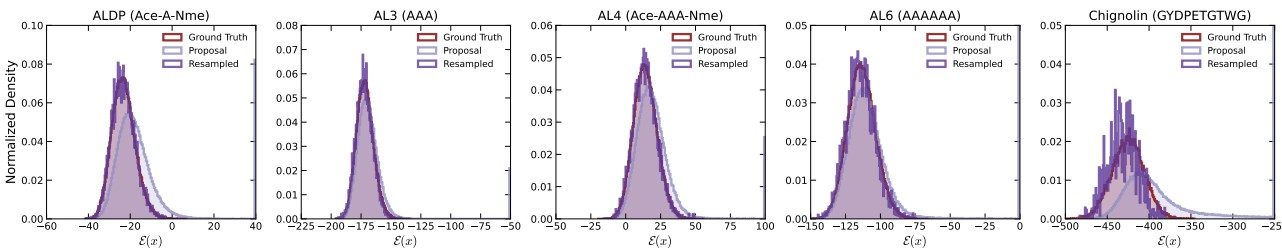

*Figure 2.* Energy histogram of the proposal and re-sampled distribution compared to ground truth MD data for all single peptide systems.

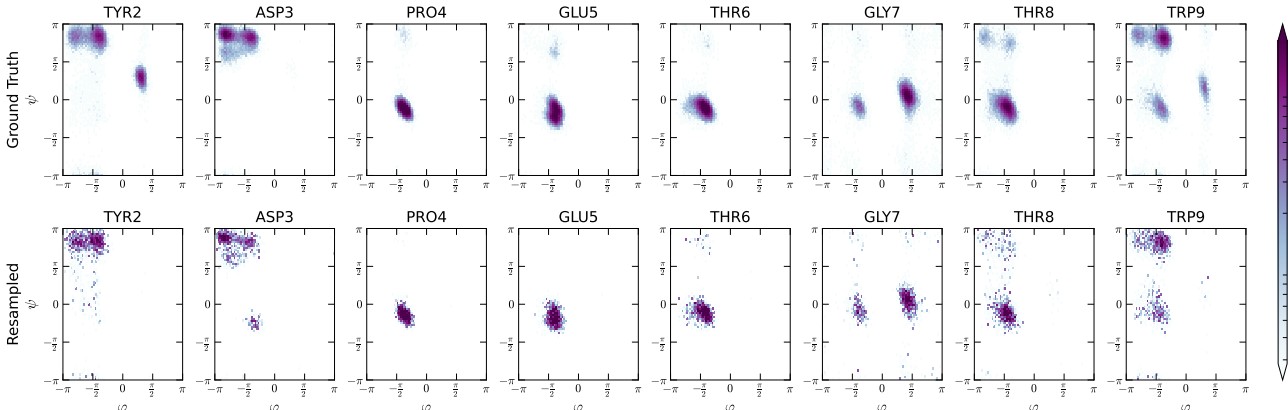

*Figure 3.* Ramachandran plots for Chignolin (top row: ground truth test set; bottom row: ARBG's predictions).

*Table 2.* Quantitative results across peptides of length 4 and 8. All methods evaluated a budget of $10^4$ energy evaluations (top) or $2 \times 10^5$ (bottom). Best values in **bold**, with second-best underlined.

| # Residues → | 4AA (30 systems) | | | 8AA (30 systems) | | |
|---|---|---|---|---|---|---|
| Model ↓ | $\mathcal{E}$-$\mathcal{W}_2$ | $\mathbb{T}$-$\mathcal{W}_2$ | TICA-$\mathcal{W}_2$ | $\mathcal{E}$-$\mathcal{W}_2$ | $\mathbb{T}$-$\mathcal{W}_2$ | TICA-$\mathcal{W}_2$ |
| TimeWarp | 7.237 | 2.204 | 0.993 | — | — | — |
| BioEmu | 90.079 | 2.037 | 1.479 | 193.873 | 4.638 | 1.601 |
| UniSim | $> 10^4$ | 2.766 | 1.733 | $> 10^3$ | 6.156 | 1.495 |
| ECNF++ | 10.032 | 1.121 | 0.572 | — | — | — |
| TarFlow | 1.260 | 0.924 | 0.492 | 11.298 | 2.733 | 1.087 |
| Prose | **0.932** | **0.752** | **0.367** | 10.038 | 2.456 | 0.988 |
| ROBIN (ours) | 1.168 | 0.886 | 0.471 | **4.251** | 2.325 | **0.943** |
| ROBIN (ours) SMC | 1.079 | 0.874 | 0.463 | 4.263 | **2.315** | 0.977 |
| $2 \times 10^5$ evals | | | | | | |
| TarFlow | 0.929 | 0.776 | 0.498 | 10.826 | 2.320 | 1.057 |
| Prose | 0.646 | **0.607** | **0.349** | 9.360 | 2.019 | 0.960 |
| ROBIN (ours) | **0.531** | 0.649 | 0.379 | **3.615** | **1.902** | **0.882** |

sequences of length 4 and 8 residues. Empirically, we observe superior performance over the current state-of-the-art method (Prose) on all molecular systems of size 8 with competitive performance on sequences of size 4.

**Twisted SMC**. We observe that Twisted SMC outperforms SNIS only marginally. We attribute this to the high quality of the base ROBIN proposal. Since the model has learned the Boltzmann distribution sufficiently well, the SMC correction yields diminishing returns. However, the twisted SMC approach confirms that ARBG is amenable to intermediate steering, paving the way for increased efficiency using early rejection, applying constraints, or guidance in larger or more complex systems where the proposal is less accurate. We find that on 8AA peptides empirically, around 7% of samples have a final partial energy greater than $100 + \mathcal{E}_{\min}$; with a 10k sample batch, where $\mathcal{E}_{\min}$ is the empirical minimum

energy obtained. For larger systems where the proposal is not as efficient, see Section 4 for additional analysis.

**Inference Scaling**. We also investigate the scaling behaviour of inference samples relative to molecular dynamics (MD) and Prose on 8AA in Figure 1 and Figure A.16. We find that ROBIN performs favourably against Prose and molecular dynamics *achieving the same performance with an order of magnitude fewer samples vs. Prose and three orders of magnitude vs. MD* in terms of $\mathbb{T}$-$\mathcal{W}_2$. Furthermore, ROBIN also outperforms Prose for the same computational budget (Figure A.16, despite operating on dimensions instead of atoms. We provide further analysis and also investigate the non-monotonic behaviour of the TICA-$\mathcal{W}_2$ for MD in § D.7. Finally, we also provide TICA plots which demonstrate the strong zero-shot performance of ROBIN on an unseen octapeptide compared to ground truth MD in Figure 4.

### 4.3. Ablations

**Performance with Bin Resolution**. As we increase the number of bins used in ARBG, the granularity of the coordinates generated by the model increases. In §B, we first discretize the coordinates into a fixed number of bins, then use uniformly sampled noise from each bin to reconstruct these molecules, demonstrating an upper bound on performance for a model with a fixed number of bins. In Figure 5 and Figure A.5, for the tri-alanine (AL3) single peptide system, we ablate the number of bins, and empirically validate that an increasing bin count monotonically improves performance on the resampled $\mathcal{E}$-$\mathcal{W}_2$.

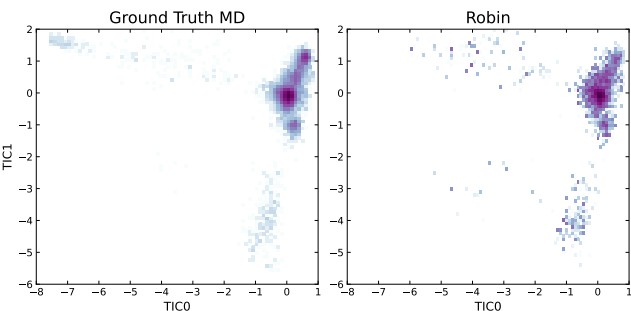

*Figure 4.* TICA plots comparing the true MD distribution against predictions from ROBIN for the octapeptide: CGSWHKQR.

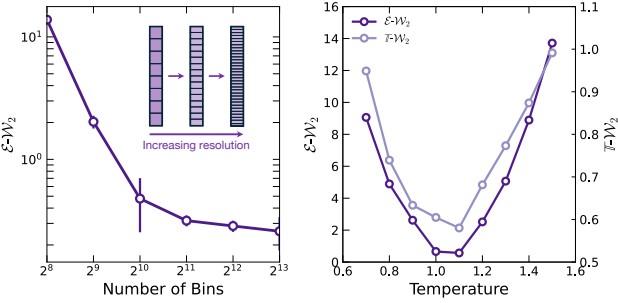

*Figure 5.* **Left**: Performance with increasing bin count for AL4. **Right**: Optimal temperature on AL4 across $\mathcal{E}\text{-}\mathcal{W}_2$ and $\mathbb{T}\text{-}\mathcal{W}_2$.

**Sampling Temperature.** We perform inference-time ablations over the transformer's sampling temperature to identify conditions that yield optimal generative performance. As shown in Figure A.5, the energy distribution varies systematically with temperature: at low temperatures, probability mass concentrates on high-likelihood (low-energy) modes, which can suppress or entirely miss other modes, while at higher temperatures the distribution flattens, encouraging diversity but potentially over-sampling modes and degrading generative quality. We quantify this trade-off using our resampled metrics via a temperature sweep, which reveals an optimal sampling temperature near $T = 1.02$ for alanine tetrapeptide. We perform equivalent temperature sweeps for all other settings and summarize the optimal temperatures in §D.1, finding that lower temperatures improve performance on larger molecular systems.

## 5. Related Work

**Boltzmann Generators.** The use of deep generative models in equilibrium sampling was popularized with the introduction of Boltzmann Generators (BGs) (Noé et al., 2019). Most subsequent work has focused on refining the NF architecture used in BGs to improve expressivity (Zhai et al., 2025; Draxler et al., 2024), stability (Schopmans & Friederich, 2025; von Klitzing et al., 2025), and generalization (Klein & Noe, 2024). CNFs offer superior expressivity and easy handling of symmetries (Köhler et al., 2020; Klein et al., 2023b), but incur a high computational

cost for likelihood evaluation during inference, which is only partially ameliorated with approximate few-step models (Rehman et al., 2026a), architectural constraints (Gloy & Olsson, 2025), or learned energy functions (Aggarwal et al., 2025; Akhound-Sadegh et al., 2025).

**Autoregressive Models on Continuous Spaces.** Several works have applied transformer-based models in generating molecular structures, including latent space (Murtada et al., 2025), flow-based (Cheng et al., 2025; Team et al., 2025). While it is possible to design an autoregressive SE(3) invariant architecture (Gebauer et al., 2019), these have not scaled as well as purely transformer-based methods. Mixture Density Networks (Bishop, 1994) have been explored for decades as a way to parametrize outputs over continuous spaces while providing exact densities (Razavi et al., 2024). These have been employed for generating sketches (Ha & Eck, 2018), images (Salimans et al., 2017), world models (Ha & Schmidhuber, 2018), and demand forecasting (Li et al., 2025), to name a few. AR MDNs have also recently been reintroduced (Tschannen et al., 2024; Li et al., 2024; Billera et al., 2024); however, all these methods focus on proposal quality, rather than interrogating the use of likelihoods for Boltzmann Generation. Moreover, MDNs have historically been prone to dropping modes in their mixtures (Deng et al., 2022), leading to suboptimal performance—as also observed for molecules in Table 1.

## 6. Conclusion

In this work, we introduced AUTOREGRESSIVE BOLTZMANN GENERATORS (ARBG), a novel autoregressive model for Boltzmann Generators that offers a promising alternative to the dominant paradigm of flow-based approaches. In particular, ARBG offers new tools that circumvent the expressivity and efficiency constraints that hinder discrete flow-based architectures while being more computationally efficient at likelihood estimation than CNFs. Importantly, ARBG enjoys the same toolkit of LLMs that comes with feature rich optimizations that enable scaling laws for language, token level-steering, which we demonstrated in the molecular setting for the first time within a Boltzmann generator framework.

**Limitations.** While ARBG enables a drastically different approach to BGs, it comes with a few notable limitations. Firstly, AR models impose a specific ordering over dimensions, while molecules themselves do not possess a natural ordering and as a result, this choice may affect performance, e.g., in small molecules (Cheng et al., 2025). Secondly, the use of uniform binning bounds the precision of the model by $\Delta$, which may pose challenges on even larger systems with sharper energy profiles. Finally, flow-based BGs can benefit from the use of informative priors, such as in TFEP (Wirnsberger et al., 2020); an investigation of which for AR models we leave as a direction for future work.

## Impact Statement

This paper presents work whose goal is to advance the field of machine learning. There are many potential societal consequences of our work, none of which we feel must be specifically highlighted here.

## Acknowledgements

The authors would like to thank Benjamin Murrell for planting the seeds of this idea as well as Luka Mucko for feedback on an early draft of this work. Danyal Rehman received financial support from the Natural Sciences and Engineering Research Council's (NSERC) Banting Postdoctoral Fellowship under Funding Reference No. 198506. The authors acknowledge funding from UNIQUE, CIFAR, NSERC, Intel, and Samsung. The research was enabled in part by computational resources provided by the Digital Research Alliance of Canada (https://alliancecan.ca), Mila (https://mila.quebec), Aithyra (https://www.oeaw.ac.at/aithyra), and NVIDIA.

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

# Appendix

## A. Theory

**Proposition 1.** *Let the true conditional density be given by $p^*(x_j|x_{<j})$ and the autoregressive model's conditional density $p_\theta(x_j|x_{<j})$ under the uniform bin parameterization. The resulting minimum achievable error of the autoregressive model in KL is,*

$$\inf_\theta \mathbb{D}_{\mathrm{KL}}\left(p^*(x_j|x_{<j})\|p_\theta(x_j|x_{<j})\right) =$$

$$\sum_{l=1}^{L} p^\star(b_l|x_{<j})\mathbb{D}_{\mathrm{KL}}\left(p^\star\left(\tilde{x}_j|\tilde{x}_j \in b_l, x_{<j}\right)\|\mathrm{Unif}(\Delta)\right).$$

*Proof.* Fix a dimension $j$ and a context $x_{<j}$. Let $\{b_l\}_{l=1}^L$ be a measurable partition of the domain of $\tilde{x}_j$ into disjoint bins, each with width $|b_l| = \Delta$, i.e. $\bigcup_{l=1}^L b_l$ covers the support of interest and $b_l \cap b_{l'} = \emptyset$ for $l \neq l'$.

Under uniform binning, the autoregressive model's conditional density is constrained to be piecewise-uniform on bins. Concretely, there exist bin probabilities $\pi_\theta(b_l|x_{<j}) \geq 0$ with $\sum_{l=1}^L \pi_\theta(b_l|x_{<j}) = 1$ such that

$$p_\theta(\tilde{x}_j|x_{<j}) = \sum_{l=1}^L \pi_\theta(b_l|x_{<j})\,\mathrm{Unif}(\Delta) = \sum_{l=1}^L \pi_\theta(b_l|x_{<j})\frac{\mathbf{1}\{\tilde{x}_j \in b_l\}}{\Delta}. \tag{5}$$

Equivalently, for $\tilde{x}_j \in b_l$,

$$p_\theta(\tilde{x}_j|x_{<j}) = \frac{\pi_\theta(b_l|x_{<j})}{\Delta}. \tag{6}$$

By the definition of the true conditionals we have,

$$\mathbb{D}_{\mathrm{KL}}(p^\star(\tilde{x}_j|x_{<j}) \| p_\theta(\tilde{x}_j|x_{<j})) = \int p^\star(\tilde{x}_j|x_{<j}) \log\left(\frac{p^\star(\tilde{x}_j|x_{<j})}{p_\theta(\tilde{x}_j|x_{<j})}\right) d\tilde{x}_j. \tag{7}$$

Since the bins form a disjoint partition, we can split the integral:

$$\mathbb{D}_{\mathrm{KL}}(p^\star\|p_\theta) = \sum_{l=1}^L \int_{b_l} p^\star(\tilde{x}_j|x_{<j}) \log\left(\frac{p^\star(\tilde{x}_j|x_{<j})}{p_\theta(\tilde{x}_j|x_{<j})}\right) d\tilde{x}_j. \tag{8}$$

Now use Eq. 6 inside each bin: for $\tilde{x}_j \in b_l$, $p_\theta(\tilde{x}_j|x_{<j}) = \pi_\theta(b_l|x_{<j})/\Delta$. Therefore

$$\log\left(\frac{p^\star(\tilde{x}_j|x_{<j})}{p_\theta(\tilde{x}_j|x_{<j})}\right) = \log p^\star(\tilde{x}_j|x_{<j}) - \log \pi_\theta(b_l|x_{<j}) + \log \Delta. \tag{9}$$

Plugging Eq. 9 into Eq. 8 yields

$$\mathbb{D}_{\mathrm{KL}}(p^\star\|p_\theta) = \sum_{l=1}^L \int_{b_l} p^\star(\tilde{x}_j|x_{<j})\left(\log p^\star(\tilde{x}_j|x_{<j}) - \log \pi_\theta(b_l|x_{<j}) + \log \Delta\right) d\tilde{x}_j$$

$$= \sum_{l=1}^L \int_{b_l} p^\star(\tilde{x}_j|x_{<j}) \log p^\star(\tilde{x}_j|x_{<j})\, d\tilde{x}_j \;-\; \sum_{l=1}^L p^\star(b_l|x_{<j}) \log \pi_\theta(b_l|x_{<j}) \;+\; \log \Delta. \tag{10}$$

(Here we used $\int_{b_l} p^\star(\tilde{x}_j|x_{<j})\, d\tilde{x}_j = p^\star(b_l|x_{<j})$ and $\sum_l p^\star(b_l|x_{<j}) = 1$.)

Now insert and subtract $\log p^\star(b_l|x_{<j})$ to isolate a discrete KL. We can then rewrite the term involving $\log \pi_\theta$ as follows:

$$-\sum_{l=1}^L p^\star(b_l|x_{<j}) \log \pi_\theta(b_l|x_{<j}) = -\sum_{l=1}^L p^\star(b_l|x_{<j}) \log p^\star(b_l|x_{<j}) + \sum_{l=1}^L p^\star(b_l|x_{<j}) \log\left(\frac{p^\star(b_l|x_{<j})}{\pi_\theta(b_l|x_{<j})}\right). \tag{11}$$

The second sum is exactly the KL divergence between the true bin-mass distribution and the model bin distribution:

$$\mathbb{D}_{\mathrm{KL}}(p^\star(b_l|x_{<j})\|\pi_\theta(b_l|x_{<j})) = \sum_{l=1}^L p^\star(b_l|x_{<j}) \log\left(\frac{p^\star(b_l|x_{<j})}{\pi_\theta(b_l|x_{<j})}\right). \tag{12}$$

Substituting back into Eq. 10 gives

$$\mathbb{D}_{\mathrm{KL}}(p^\star\|p_\theta) = \mathbb{D}_{\mathrm{KL}}(p^\star(b_l|x_{<j})\|\pi_\theta(b_l|x_{<j})) + \sum_{l=1}^{L}\left[\int_{b_l} p^\star(\tilde{x}_j|x_{<j})\log\left(\frac{p^\star(\tilde{x}_j|x_{<j})}{p^\star(b_l|x_{<j})/\Delta}\right)d\tilde{x}_j\right]. \qquad (13)$$

To interpret the second term, observe that for $\tilde{x}_j \in b_l$,

$$\frac{p^\star(b_l|x_{<j})}{\Delta} = p^\star(b_l|x_{<j})\cdot\mathrm{Unif}(\Delta).$$

Moreover, using the true conditional density,

$$p^\star(\tilde{x}_j|\tilde{x}_j \in b_l, x_{<j}) = \frac{p^\star(\tilde{x}_j|x_{<j})}{p^\star(b_l|x_{<j})}\mathbf{1}\{\tilde{x}_j \in b_l\}.$$

Hence, we can rewrite the bracketed integral as

$$\int_{b_l} p^\star(\tilde{x}_j|x_{<j})\log\left(\frac{p^\star(\tilde{x}_j|x_{<j})}{p^\star(b_l|x_{<j})/\Delta}\right)d\tilde{x}_j = p^\star(b_l|x_{<j})\int_{b_l} p^\star(\tilde{x}_j|\tilde{x}_j \in b_l, x_{<j})\log\left(\frac{p^\star(\tilde{x}_j|\tilde{x}_j \in b_l, x_{<j})}{\mathrm{Unif}(\Delta)}\right)d\tilde{x}_j$$

$$= p^\star(b_l|x_{<j})\mathbb{D}_{\mathrm{KL}}(p^\star(\tilde{x}_j|\tilde{x}_j \in b_l, x_{<j})\|\mathrm{Unif}(\Delta)). \qquad (14)$$

Combining Eq. 13 and Eq. 14, we obtain the exact decomposition

$$\mathbb{D}_{\mathrm{KL}}(p^\star(\tilde{x}_j|x_{<j})\|p_\theta(\tilde{x}_j|x_{<j})) = \mathbb{D}_{\mathrm{KL}}(p^\star(b_l|x_{<j})\|\pi_\theta(b_l|x_{<j})) + \sum_{l=1}^{L}p^\star(b_l|x_{<j})\mathbb{D}_{\mathrm{KL}}(p^\star(\tilde{x}_j|\tilde{x}_j \in b_l, x_{<j})\|\mathrm{Unif}(\Delta)).$$

$$(15)$$

The second term in Eq. 15 depends only on the true distribution $p^\star$ and the fixed bins, and is independent of $\theta$. The first term is a KL divergence over discrete distributions, hence it is always nonnegative and is minimized if and only if

$$\pi_\theta(b_l|x_{<j}) = p^\star(b_l|x_{<j}) \quad \text{for all } l.$$

At this optimum, the discrete KL equals zero, so the minimum achievable KL within the piecewise-uniform model family is

$$\inf_\theta \mathbb{D}_{\mathrm{KL}}(p^\star(\tilde{x}_j|x_{<j})\|p_\theta(\tilde{x}_j|x_{<j})) = \sum_{l=1}^{L}p^\star(b_l|x_{<j})\mathbb{D}_{\mathrm{KL}}(p^\star(\tilde{x}_j|\tilde{x}_j \in b_l, x_{<j})\|\mathrm{Unif}(\Delta)).$$

As a result, this matches the statement of the proposition. $\qquad\square$

## B. Data Preprocessing and Analysis

We analyze the coordinate distribution of each peptide by placing the training data into a fixed number of bins (set to num_bins = 1024) in Figure A.1.

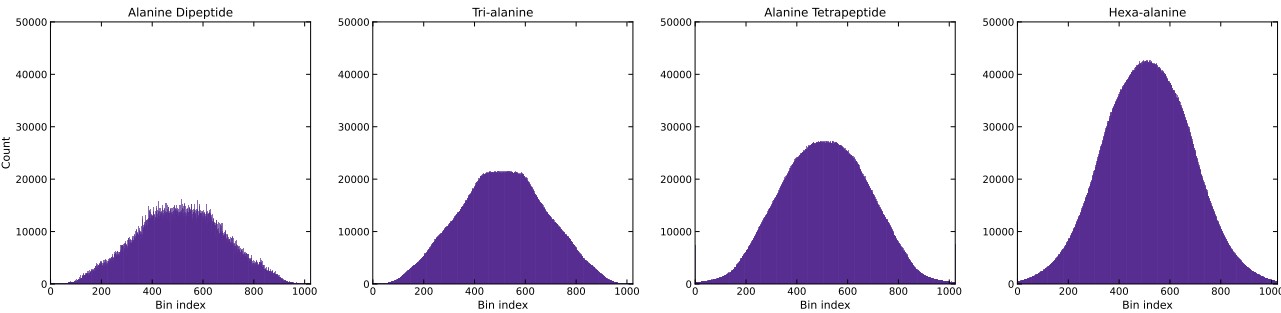

*Figure A.1.* The distribution of coordinates across bins for fixed bin count with num_bins = 1024.

In addition to analyzing the distribution of samples across discretized bins, we evaluated a lower bound on $\mathcal{E}\text{-}\mathcal{W}_2$ and $\mathbb{T}\text{-}\mathcal{W}_2$ by de-quantizing via uniform noise injection (see Figure A.2). Specifically, training data were discretized into bins and subsequently mapped back to continuous coordinates by reconstructing via uniformly sampled noise. This procedure was repeated over a range of bin sizes to characterize the bin-width–dependent lower bound on attainable performance across

all alanine-based peptides. We further conducted ablations to assess the effect of bin size on $\mathcal{E}\text{-}\mathcal{W}_2$ and $\mathbb{T}\text{-}\mathcal{W}_2$. As shown in Figure A.3, performance exhibits a near-monotonic dependence on bin resolution, validating our original analysis. Based on these results, we used 4096 bins for alanine dipeptide and tri-alanine, and 8192 bins for larger molecules and ROBIN.

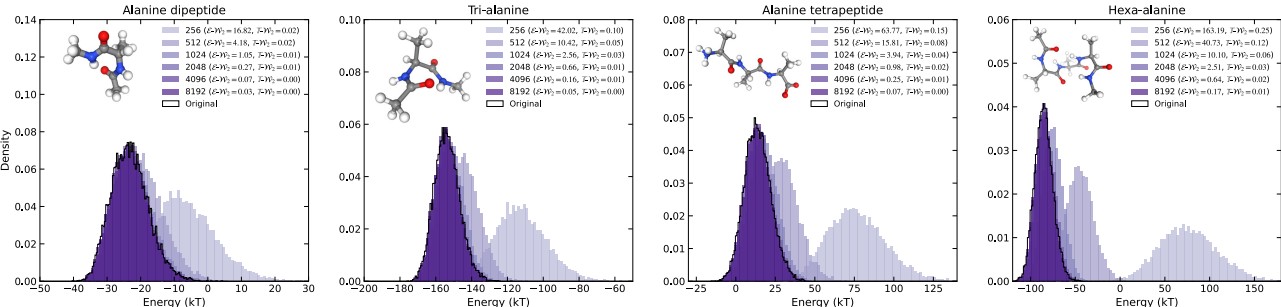

*Figure A.2.* Energy distributions on the training data as a function of bin discretization. The corresponding $\mathcal{E}\text{-}\mathcal{W}_2$ and $\mathbb{T}\text{-}\mathcal{W}_2$ values are reported for each discretization, demonstrating an upper bound on the learnability of these metrics.

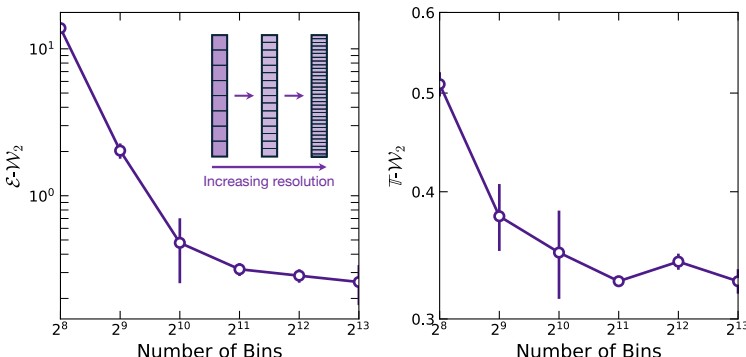

*Figure A.3.* We vary the model's bin count and evaluate its impact on the resampled $\mathcal{E}\text{-}\mathcal{W}_2$ and $\mathbb{T}\text{-}\mathcal{W}_2$ for tri-alanine.

## C. Metrics

Below, we introduce the metrics used to evaluate model performance and describe their computation. The proposed metrics capture both local and global behaviour. Energy-based metrics assess the accuracy of local interactions, as small geometric perturbations can induce large energy variations. Complementary global metrics—including torus- and TICA-based measures—evaluate mode coverage and the ability of models to capture multi-modal structure. We omit the effective sample size (ESS), as its interpretation is invalidated by the use of SMC.

### C.1. Main Geometric Metrics

**2-Wasserstein Energy Distance** ($\mathcal{E}\text{-}\mathcal{W}_2$). To quantify the agreement between generated and reference energy distributions, we compute the squared 2-Wasserstein distance between the energies of generated samples and those obtained from MD. Let $p, q \in \mathcal{P}(\mathbb{R})$ denote the probability distributions over energy values for the generated and reference samples, respectively, and let $\Pi(p, q)$ denote the set of admissible couplings between them. The Wasserstein energy distance is then defined as:

$$\mathcal{E}\text{-}\mathcal{W}_2(p, q)^2 \;\triangleq\; \min_{\pi \in \Pi(p,q)} \int_{\mathbb{R} \times \mathbb{R}} |x - y|^2 \, \mathrm{d}\pi(x, y). \tag{16}$$

This metric measures how closely the generated energy landscape matches the reference distribution. Because molecular energies are highly sensitive to local structural change such as bond lengths/angles, $\mathcal{E}\text{-}\mathcal{W}_2$ is particularly effective at detecting physically relevant discrepancies. Lower values correspond to better agreement with the target Boltzmann distribution.

**Torus 2-Wasserstein Distance** ($\mathbb{T}\text{-}\mathcal{W}_2$). To assess structural similarity in torsional space, we compute a 2-Wasserstein distance defined on the torus. For a molecule with $L \in \mathbb{N}$ residues, each conformation is represented by its vector of

dihedral angles:

$$\text{Dihedrals}(x) = (\phi_1, \psi_1, \ldots, \phi_{L-1}, \psi_{L-1}) \in [0, 2\pi)^{2(L-1)}. \tag{17}$$

To account for the periodicity of angular variables, the squared cost between two conformations $x$ and $y$ is defined as:

$$c_{\mathbb{T}}(x, y)^2 = \sum_{i=1}^{2(L-1)} \left[ \left( \text{Dihedrals}(x)_i - \text{Dihedrals}(y)_i + \pi \right) \bmod 2\pi - \pi \right]^2. \tag{18}$$

The corresponding torus Wasserstein distance between two distributions $p, q \in \mathcal{P}([0, 2\pi)^{2(L-1)})$ is then defined as:

$$\mathcal{T}\text{-}\mathcal{W}_2(p, q)^2 \triangleq \min_{\pi \in \Pi(p,q)} \int c_{\mathbb{T}}(x, y)^2 \, d\pi(x, y). \tag{19}$$

This metric captures global conformational differences in torsional space while respecting angular periodicity. Unlike energy-based distances, $\mathbb{T}\text{-}\mathcal{W}_2$ is sensitive to missing or misrepresented conformational modes, providing a complementary assessment of structural diversity and coverage in generative Boltzmann models. One point of note is that although this claim generally holds, in cases where there are few samples from a given mode that are lost, this does not substantially impact the $\mathbb{T}\text{-}\mathcal{W}_2$, meaning that we can see a reduced value even in the presence of mode loss—one clear example of this phenomenon is presented in Section D.3, where we demonstrate mode collapse despite decreasing $\mathbb{T}\text{-}\mathcal{W}_2$.

**TICA 2-Wasserstein Distance** (TICA-$\mathcal{W}_2$). To compare the long-timescale dynamical structure of trajectories, we evaluate discrepancies in a reduced space defined by time-lagged independent component analysis (TICA). TICA identifies collective coordinates that maximize autocorrelation, isolating the slow modes governing conformational dynamics.

Given a mean-centered time series $\{\tilde{x}_t\}_{t=1}^T \subset \mathbb{R}^n$ and a lag time $\tau$, we estimate the empirical covariance matrices:

$$\hat{C}_{00} = \frac{1}{T-\tau} \sum_{t=1}^{T-\tau} \tilde{x}_t \tilde{x}_t^\top, \qquad \hat{C}_{0\tau} = \frac{1}{T-\tau} \sum_{t=1}^{T-\tau} \tilde{x}_t \tilde{x}_{t+\tau}^\top. \tag{20}$$

The dominant slow modes are obtained by solving the generalized eigenvalue problem

$$\hat{C}_{0\tau} w = \lambda \hat{C}_{00} w, \tag{21}$$

where each eigenvector $w$ defines a linear projection with maximal normalized autocorrelation at lag $\tau$. In practice, we retain the first two TICA components $\{w_1, w_2\}$, which capture the slowest dynamical processes.

Using these projections, we define an $\ell_2$ cost between configurations $x, y \in \mathbb{R}^n$ as their Euclidean distance in TICA space:

$$c_{\text{TICA}}(x, y)^2 = \sum_{j=1}^2 \left( w_j^\top x - w_j^\top y \right)^2. \tag{22}$$

The corresponding TICA Wasserstein distance between generated and reference distributions $p, q \in \mathcal{P}(\mathbb{R}^n)$ is then

$$\text{TICA-}\mathcal{W}_2(p, q)^2 \triangleq \min_{\pi \in \Pi(p,q)} \int c_{\text{TICA}}(x, y)^2 \, d\pi(x, y). \tag{23}$$

This metric directly assesses agreement in the slow dynamical subspace learned from the reference trajectory. By construction, TICA-$\mathcal{W}_2$ is sensitive to mismatches in metastable state populations and transition pathways, making it well-suited for evaluating models intended to reproduce long-timescale molecular kinetics.

### C.2. On the use of Geometric over Likelihood-based metrics in High-dimensions

In this work, we prioritize Wasserstein-based metrics (TICA, Torus, Energy) over likelihood-based metrics. While ESS is a standard diagnostic for the efficiency of SNIS estimators, it is widely recognized as a potentially misleading proxy for sample quality in high-dimensional spaces. In high-dimensional spaces, importance sampling is susceptible to the "curse of dimensionality", referred to as *weight collapse* in the particle filtering literature (Snyder et al., 2008). As the dimensionality increases, the overlap between the typical sets of the proposal and target distributions vanishes exponentially. Consequently, the variance of the importance weights becomes dominated by rare samples that land in the small region of overlap. This results in an estimator variance that explodes, rendering ESS an unreliable metric for performance in systems with hundreds of degrees of freedom (e.g., Decapeptides), as the metric becomes sensitive to global scaling factors rather than local mode coverage.

**The Bias Toward Mode Collapse**. The core limitation of ESS in the context of Boltzmann Generation is its tendency to reward "mode-seeking" behaviour over "mass-covering" behaviour. ESS is derived from the variance of the importance weights $w_i(x) = \mu_{\text{target}}(x_i)/p_\theta(x_i)$. Specifically,

$$\text{ESS}\left(\{w_i\}_{i=1}^N\right) = \frac{\left(\sum_{i=1}^N w_i\right)^2}{N \sum_{i=1}^N w_i^2}.$$

where $x_i \sim p_\theta$. A generative model $p_\theta$ can maximize ESS by collapsing its probability mass into a single, highly stable metastable state (a single mode of $\mu_{\text{target}}$). In this scenario, the ratio $\mu_{\text{target}}(x)/p_\theta(x)$ remains stable within that specific region, yielding a high ESS; however, this comes at the cost of failing to sample other metastable states (mode dropping). In Figure A.4, we compare the Ramachandran plots between the ground truth MD data, FALCON, and ARBG for one of the torsion angles present in tri-alanine. It can clearly be seen that the mode between 0 and $\frac{\pi}{2}$ exists in the training data, while being lost in FALCON—a model that obtains a higher ESS than ARBG.

Conversely, a model that attempts to cover the full diversity of the Boltzmann distribution ("mass-covering") is much more likely to assign non-zero probability to high-energy regions where $\mu_{\text{target}}(x) \approx 0$. This results in high variance of the importance weights and a low ESS, despite the model being superior in terms of exploring the global conformational space.

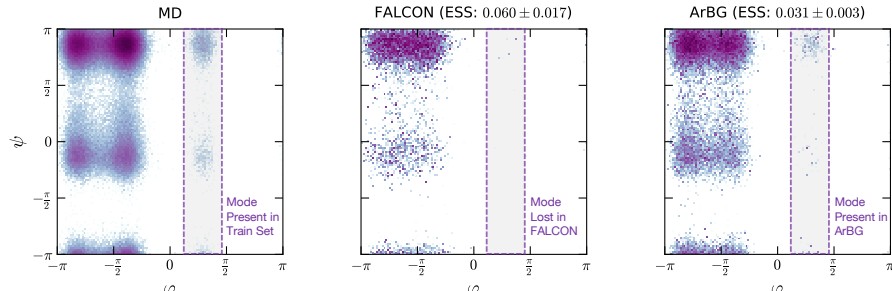

*Figure A.4.* **Left**: Ramachandran plot from the ground truth MD data for tri-alanine; **Center**: FALCON's torsion angle predictions; **Right**: Torsion angle predictions from ARBG. FALCON clearly loses one of the conformational modes at inference.

**On the Interaction with Numerical Error in Practice**. In practice, with models that have some numerical error, the variance of importance weights—and therefore ESS—is often dominated by numerical errors. Where a small fraction of samples will compute as having an unusually high likelihood. This causes the creation of a large importance weight. This is why, in practice, all models utilize a form of clipping to ensure reasonable importance weight values and to prevent collapse. In this work, we use a clipping value of $0.002$ where the samples with the largest $0.002$ fraction of importance weights are clipped following prior work (Klein et al., 2023b; Midgley et al., 2023; Tan et al., 2025b; Rehman et al., 2026a). In practice, ESS is highly sensitive to numerical precision and is often dominated by outliers, particularly in high-dimensional settings. By contrast, geometry-based metrics are substantially more robust to such numerical effects and provide a more reliable characterization of global model behaviour.

## D. Additional Results

### D.1. Temperature Tuning

Sampling temperature controls the entropy of the model distribution by scaling logits at inference time. The optimal temperatures for all systems are reported in Table A.1. For smaller systems, temperatures near 1.0 are sufficient, with slight gains observed for values marginally above 1.0. In contrast, larger and more complex systems benefit from lower temperatures, which likely mitigate underfitting.

**Temperature on Alanine Tetrapeptide**. In Figure A.5, we study the effects of temperature tuning on the proposal and re-weighted distributions on the alanine tetrapeptide system. First, as noted in the main text, we find that the optimal temperature is slightly higher than 1.0. This is interesting and in line with prior works that find a slightly more diffuse proposal may be slightly better for Boltzmann Generation metrics, as it allows better coverage of the space.

*Table A.1.* Optimal sampling temperatures identified via inference-time temperature sweeps across molecular systems.

| System | Optimal Temperature, $T$ |
|---|---|
| Alanine Dipeptide | 1.03 |
| Tri-alanine | 0.99 |
| Alanine Tetrapeptide | 1.02 |
| Hexa-alanine | 0.98 |
| Chignolin | 0.88 |
| ROBIN (Transferable) | 0.95 |

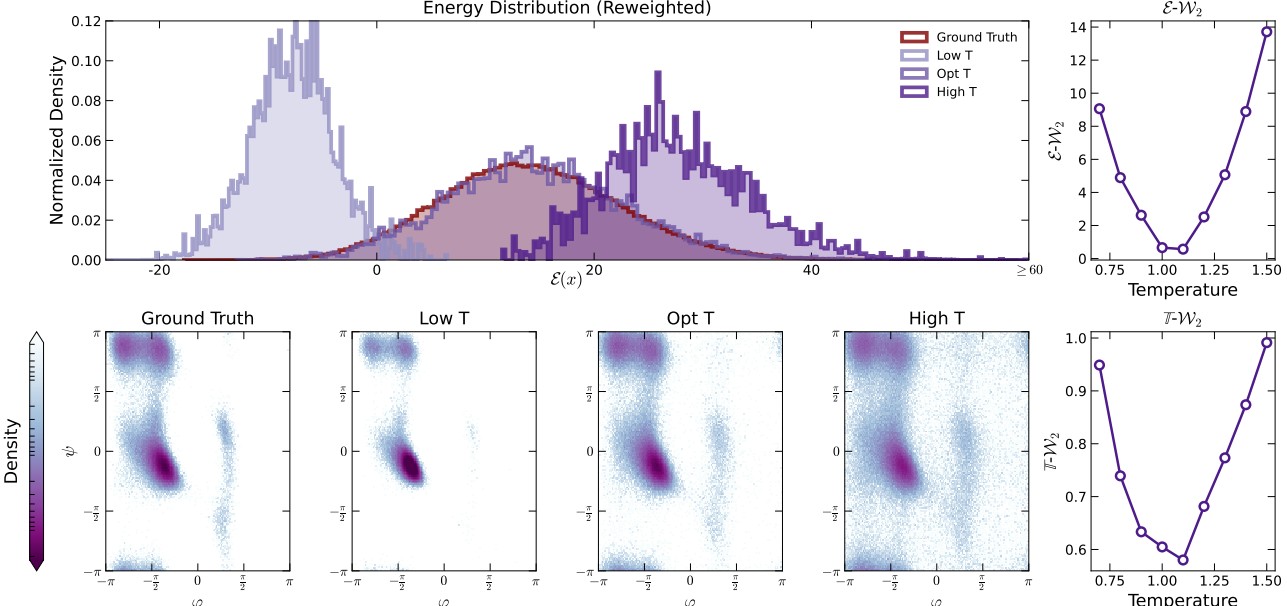

*Figure A.5.* Ablations on model temperature. For the energy distribution, we demonstrate that lower temperatures sample lower energy modes more frequently, while the converse holds for higher temperatures. We also show how the modes become more prominent at high temperatures and are lost at lower temperatures. Finally, we show how an optimal temperature exists for optimizing $\mathcal{E}\text{-}\mathcal{W}_2$ and $\mathbb{T}\text{-}\mathcal{W}_2$.

### D.2. Inference Time

In Table A.2, we report inference throughput (samples per second) for transferable models with 2, 4, and 8 residues, averaged over 30 systems. While ROBIN is slower than Prose, its substantially higher sample quality yields superior performance under a fixed sampling budget (see Figure 1).

### D.3. Alanine Dipeptide

Below, we summarize the results of all ARBG variants in conjunction with other baselines on ALDP. ARBG outperforms all competing models—including both discrete flows and CNFs, on $\mathcal{E}\text{-}\mathcal{W}_2$, with competitive performance on $\mathbb{T}\text{-}\mathcal{W}_2$.

**Mode Collapse**. When training models on the alanine dipeptide dataset from Klein et al. (2023b), we observe that we can continue improving our performance across both global and local metrics if we train our models for longer; however, in this process, part of the performance improvement comes from losing the mode, which artificially inflates ESS. Torus, which is designed to be a global metric, also suffers given that there are an insufficient number of points in that mode to radically impact the degradation of performance, yielding a nearly monotonic trend in performance improvement as training time increases. For larger systems, like tri-alanine and above, this behaviour is not observed.

In Figure A.6, we provide a clear demonstration of the lost mode on ALDP. The Ramachandran plots are shown for two different instances in the training process—Epoch 110 and Epoch 370. We show that earlier in training, the mode exists, but as training continues, it disappears. This can also be observed on the training loss curve as annotated. We also provide the ESS and Torus results during training to demonstrate that the loss of the mode improves performance on metrics.

*Table A.2.* Inference speed in samples per second for best performing models in the transferable setting. ROBIN is around 50% faster than Prose per model evaluation, but is slower in terms of samples per second due to operating over dimensions instead of atom coordinates and therefore requires $3\times$ the model evaluations.

|  | 2AA | 4AA | 8AA |
|---|---|---|---|
| TarFlow | 737 | 329 | 126 |
| PROSE | 338 | 158 | 66 |
| ROBIN | 260 | 87 | 29 |

*Table A.3.* Results on alanine dipeptide. Best results are **bolded**, with second-best underlined.

| | Alanine dipeptide (ALDP) | |
|---|---|---|
| Algorithm ↓ | $\mathcal{E}$-$\mathcal{W}_2 \downarrow$ | $\mathbb{T}$-$\mathcal{W}_2 \downarrow$ |
| BoltzNCE | $0.27 \pm 0.02$ | $0.57 \pm 0.00$ |
| SE(3)-EACF | 108.202 | 2.867 |
| ECNF | 0.419 | 0.311 |
| RegFlow | $0.501 \pm 0.011$ | $0.951 \pm 0.054$ |
| ECNF++ | $0.914 \pm 0.122$ | $0.189 \pm 0.019$ |
| SBG | $0.741 \pm 0.189$ | $0.431 \pm 0.141$ |
| FALCON-A | $0.512 \pm 0.038$ | $\underline{0.180 \pm 0.005}$ |
| FALCON | $\underline{0.225 \pm 0.104}$ | $0.402 \pm 0.021$ |
| GIVT | $0.256 \pm 0.033$ | $\mathbf{0.175 \pm 0.171}$ |
| MoL-PixelCNN++ | $1.447 \pm 0.277$ | $0.528 \pm 0.028$ |
| GMM-PixelCNN++ | $0.763 \pm 0.118$ | $0.354 \pm 0.098$ |
| ARBG | $\mathbf{0.209 \pm 0.041}$ | $0.402 \pm 0.008$ |

We believe this stems from the method used to generate the dataset in Klein et al. (2023b). Specifically, the dataset was generated in the following way:

1. MD simulation using *Amberff99SBildn* force-field at 300K for 1 ms using openMM (Eastman et al., 2024) with a timestep of 1 femto-second.

2. Relaxation of $10^5$ uniformly randomly selected states from the MD data for 100 femto-seconds each using the *GFN2-xTB* forcefield (Bannwarth et al., 2018) and the ASE library (Hjorth Larsen et al., 2017) with a friction constant of 0.5 a.u.

3. To make the density nearly equal between negative and positive $\varphi$ dihedral angles, importance sampling is performed using weights from a von Mises distribution $f_{vM}$. Specifically, weights for each sample are computed as:

$$\omega(\varphi) = 150 f_{vM}(\varphi | \mu = 1, \kappa = 10) + 1 \tag{24}$$

with $10^5$ training samples drawn from the weighted distribution.

Specifically, this final reweighting step makes it possible for powerful models to overfit on the positive $\varphi$ mode. The reweighting step causes there to be multiple instances of exactly the same data sample in the training set. For powerful models, seeing the same datapoint multiple times (even with data augmentation) causes overfitting. We observe that the likelihood of these exact training samples explodes, causing the distribution after importance sampling to remove the positive $\varphi$ mode.

This mixed energy function usage creates somewhat of a problem for importance sampling. In practice, following previous work, use the *Amberff99SBildn* force-field at 300K as a target energy function for reweighting, but note that this is not quite a perfect fit as the samples are relaxed slightly with the *GFN2-xTB* forcefield which may create a slight mismatch between the target distribution and the actual *Amberff99SBildn*-defined equilibrium distribution.

**Recommendation**. For newer and more powerful Boltzmann Generator models, we recommend using training sets without importance sampling, as these are much more difficult to overfit on specific training samples. It is important to be mindful of overfitting-type behaviour on these small datasets with relatively powerful models.

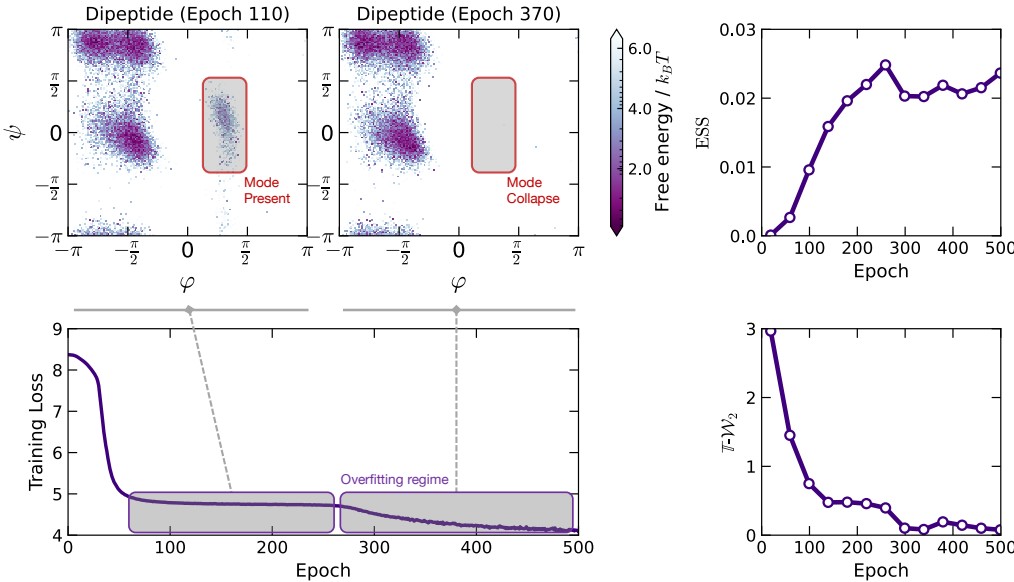

*Figure A.6.* We demonstrate that training models for too long leads to overfitting on the training data, which despite improving resampled metrics, yields undesirable behaviour.

## D.4. Ramachandran Plots for Other Single Peptide Systems

Here, we demonstrate the competitive performance of ARBG across single peptide systems by showing the Ramachandran plots for all systems considered. In all cases considered, ARBG captures nearly every mode present in the test data, clearly illustrating the quality of the learned likelihoods and their synergy with SNIS.

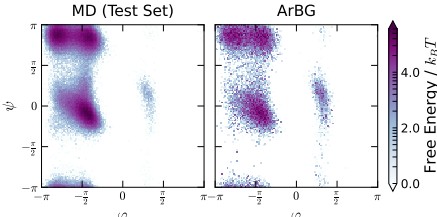

*Figure A.7.* **Left**: Test data for alanine dipeptide; **Right**: ARBG's angular predictions for alanine dipeptide.

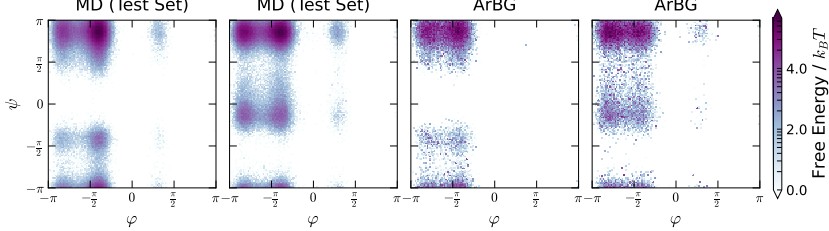

*Figure A.8.* **Left**: Test data for tri-alanine; **Right**: ARBG's angular predictions for tri-alanine.

## D.5. De-quantization Strategies

For our discrete model to generate continuous coordinates, we explored three different sampling strategies:

1. Sampling uniformly from the discrete bin selected by the model. This is our reasonable default choice and defines a piecewise-constant continuous density in $\mathbb{R}^d$.

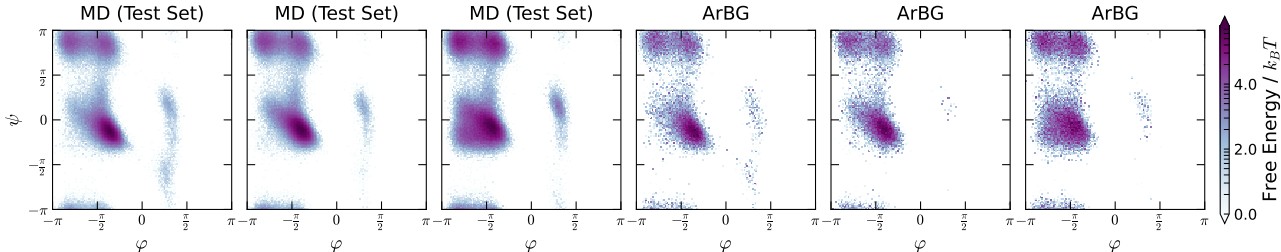

*Figure A.9.* **Left**: Test data for alanine tetrapeptide; **Right**: ARBG's angular predictions for alanine tetrapeptide.

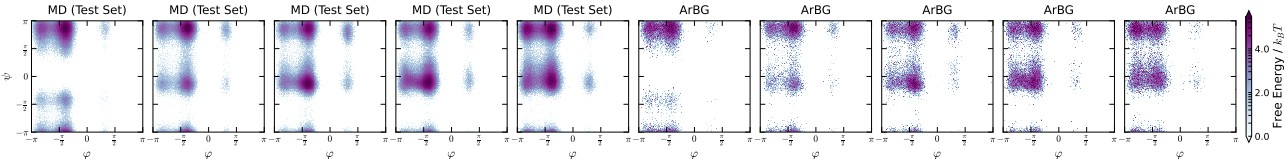

*Figure A.10.* **Left**: Test data for hexa-alanine; **Right**: ARBG's angular predictions for hexa-alanine.

2. Using the center value of the bin. This strategy reduces a bit of variability from generation and may make bond lengths slightly more uniform. With enough bins, this is not necessary. Empirically, we observed small benefits for this de-quantization strategy over uniformly sampling from the bin.

3. Using the biased training data distribution for the chosen molecule to determine empirical offsets that we apply to the chosen bin at inference time. Especially for larger bin sizes, we may be able to fit more interesting distributions concerning the training dataset within the bin. Here, we use a Dirac distribution over the empirical mean of the training dataset within the bin. We find this gives a small boost in performance, especially when the training data is centered.

### D.6. Transferable Generation

**Autoregressive Twisted SMC Efficiency Benefits**. We evaluate the efficiency gains that can be obtained using our Autoregressive Twisted SMC algorithm. The main idea is that it is easy to detect samples that will not result in valid low energy samples early on during inference. Using the twist function defined in Eq. 4, we are able to essentially stop inference early for any sample that exhibits a high partial energy. In Figure A.11, we investigate the partial energy distributions on the sequence SQQKVAFE 8AA test set peptide for ROBIN. To investigate this, we perform SMC inference without resampling for 10,000 generations. We record the partial energy of each sample at each residue checkpoint. We find that residue 2 has around 2% of samples that have poor energy samples while residue 7 has $> 7\%$ of samples that have high energies and represent likely steric clashes or other high energy features. These samples represent "wasted" compute, in that they will not contribute to the final distribution of samples. Therefore, additional efficiency can be gained by filtering these out early. On this peptide, using the $min(\mathcal{E}(x)) + 100$ filter on energy at the earliest time a sample is registered as high energy, we find a savings of roughly 3% over a method without intermediate resampling.

While this is a relatively minor saving, we expect that for more complicated and larger systems where the proposal has more failure modes, the advantage of autoregressive twisted SMC here would increase substantially in terms of cost savings or sampling efficiency, depending on the exact method and utilization of this concept.

**TICA Plots for Unseen Octapeptides**. To demonstrate the quality of ROBIN and its zero-shot performance, we include TICA plots for seven different unseen octapeptides in Figure A.12. In this process, we show the predictive capacity of ROBIN as it nearly perfectly captures all the modes of these unseen molecules.

**Peptide-level Performance and Learnability**. We evaluate the learned model on each peptide in the test set and report the resampled $\mathcal{E}\text{-}\mathcal{W}_2$ and $\mathbb{T}\text{-}\mathcal{W}_2$ in Figures A.13 and A.14 for ROBIN, Prose, and TarFlow. Overall, performance is broadly comparable across models on a per-sequence basis; however, several peptides remain challenging for all methods. For example, all models perform poorly on the $\mathcal{E}\text{-}\mathcal{W}_2$ for KRRGFFLE. Further analysis indicates that sequences with substantial charge contributions are particularly difficult to learn, especially those containing R and Y residues. Although all amino acids incur steep energy penalties outside favourable conformations, certain side chains are exceptionally sensitive to small geometric perturbations. Arginine's planar, highly charged guanidinium group exhibits strongly orientation-dependent

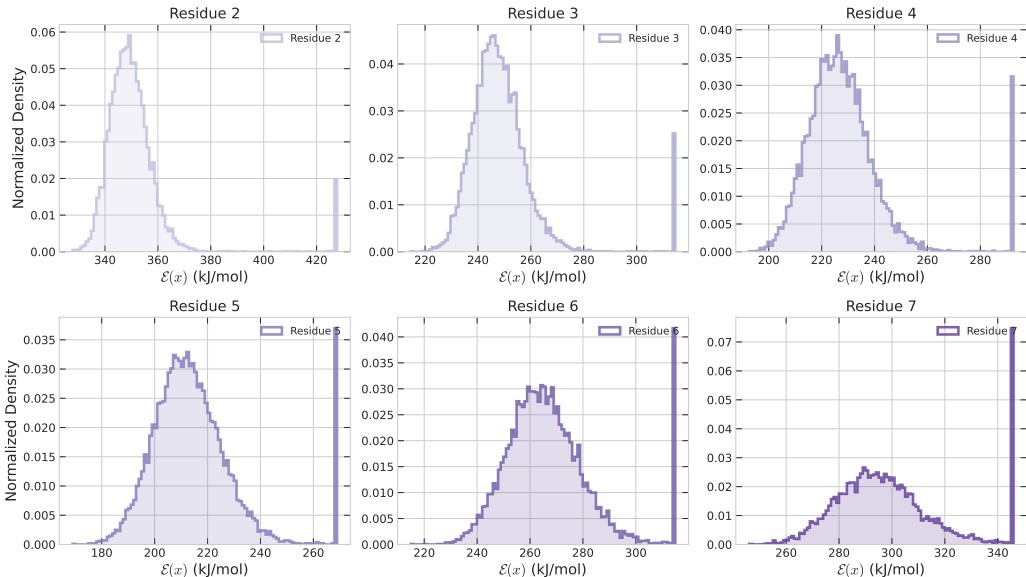

*Figure A.11.* Histogram of the energy values for 10,000 samples on SQQKVAFE for each residue, where we perform resampling for SMC. We clip the maximum energy to $\min(\mathcal{E}(x)) + 100$. The spikes represent all values greater than or equal to that histogram value. We can see that by residue 7, $> 7\%$ of samples have extremely large energies, which likely represent clashes or erroneous bond lengths.

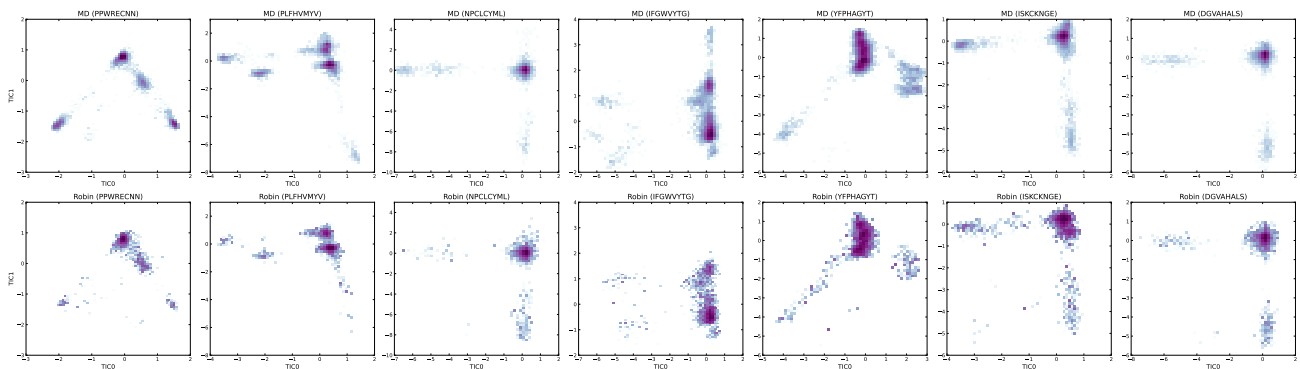

*Figure A.12.* TICA modes using ROBIN on seven unseen octapeptides (from left to right): PPWRECNN, PLFHVMYV, NPCLCYML, IFGWVYTG, YFPHAGYT, ISKCKNGE, DGVAHALS.

electrostatics, while tyrosine's aromatic ring engages in highly directional non-bonded interactions (Yoo & Aksimentiev, 2016; Jing et al., 2019). Consequently, minor deviations in side-chain geometry can produce large energy fluctuations, complicating the learning of equilibrium conformational distributions for these residues.

### D.7. Inference Scaling

In Figure 1, we demonstrated the scaling performance of ROBIN against Prose and MD on unseen octapeptide systems in terms of the number of function and energy evaluations. In this section, we continue an investigation into the inference scaling behaviour of ROBIN on octapeptides.

**GPU hour performance**. In Figure A.16, we investigate not only an equal number of energy evaluations, but also the scaling performance in terms of GPU hours. While ROBIN is slower than Prose or MD per model or energy function evaluation, its performance for the same number of GPU hours is better, especially on the $\mathbb{T}\text{-}\mathcal{W}_2$.

Here, and in Figure 1, we notice an unexpected trend in the TICA-$\mathcal{W}_2$ plots. Specifically, the TICA-$\mathcal{W}_2$ is relatively flat for MD, then spikes at around $10^7$-$10^8$ energy evaluations. We investigate this further by breaking the performance out sequence by sequence in Figures A.17 to A.19, where we show all 30 test set eight residue sequences. We notice a few interesting things, as stated in the following:

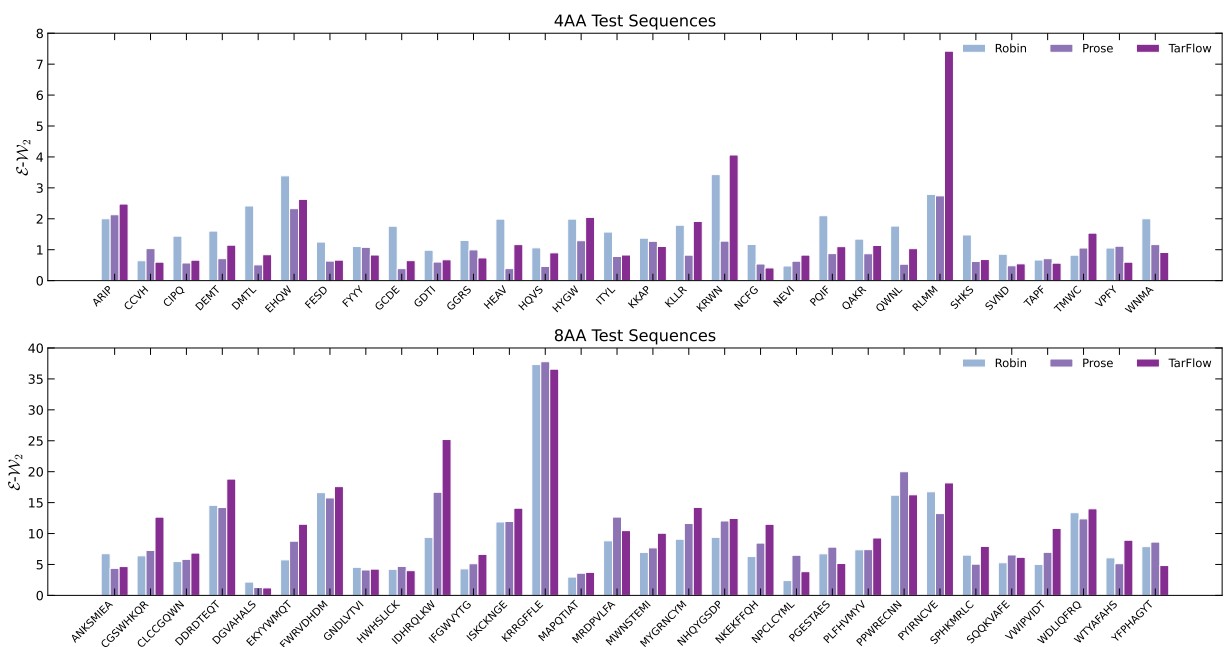

*Figure A.13.* The resampled $\mathcal{E}$-$\mathcal{W}_2$ across peptides when comparing ROBIN, Prose, and SBG. Models were evaluated using $10^4$ samples.

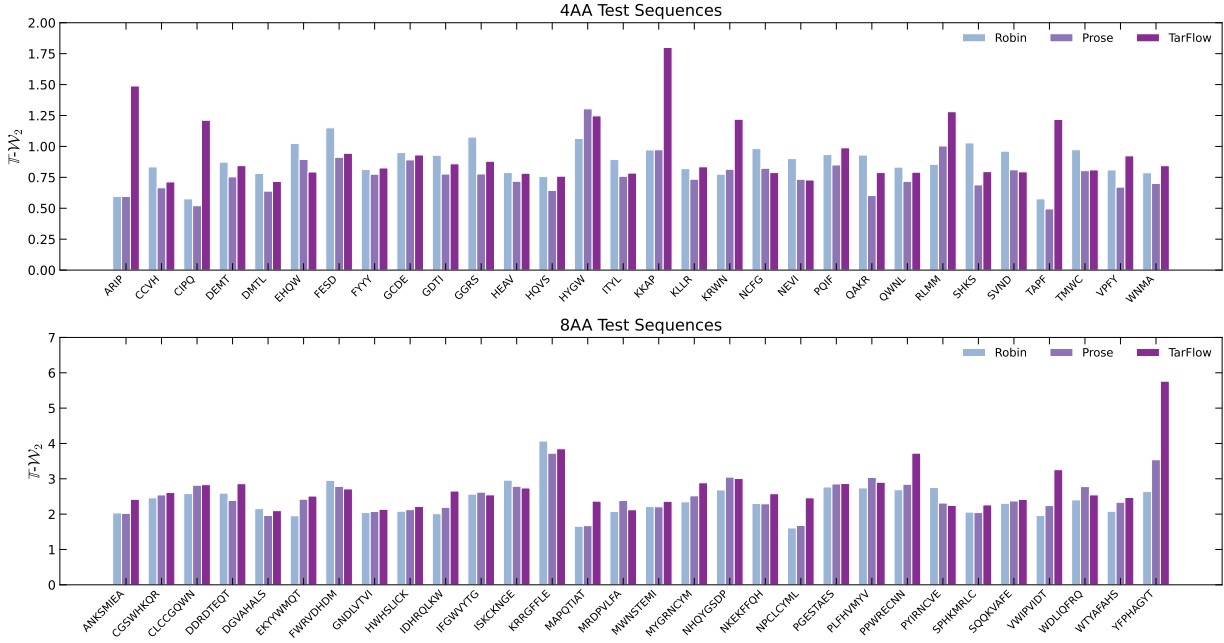

*Figure A.14.* The resampled $\mathbb{T}$-$\mathcal{W}_2$ across peptides when comparing ROBIN, Prose, and SBG. Models were evaluated using $10^4$ samples.

1. As noted in Figures A.13 and A.14, the variability between sequences is quite high. Often, the performance is non-monotonic. For some sequences, one model is better than another, particularly at low energy evaluations; however, as the number of energy evaluations grows, ROBIN generally outperforms Prose.

2. TICA-$\mathcal{W}_2$ for MD often has extremely non-monotonic elements often after $10^7$ energy evaluations due to mode jumps. Specifically, it takes around $10^7$ MD steps for chains to jump to the next mode. This often drastically changes TICA-$\mathcal{W}_2$ as the relative weights between modes evolve, especially when the new mode has less free energy than the starting mode. We demonstrate this clearly in Figure A.15, where we see the MD over-sample a mode that is incorrectly weighted,

leading to significant spikes in the TICA-$\mathcal{W}_2$.

We note that these plots are compared to the test set trajectories which have been run for 50 times as long as the longest MD chain. Given that we see the first mode mixing events at around $10^7$ energy evaluations, this implies that the test chains may not be fully mixed.

An interesting difference between MD and BG traces is that BG traces are often more monotonic than the MD trajectories. This is reasonable as BGs (both Prose and ROBIN) sample independently from their proposal where MD is autocorrelated. This means that the performance is often significantly better for BG type models in the few-step regime for unseen peptides. ROBIN outperforms all others on 8 residue systems on average.

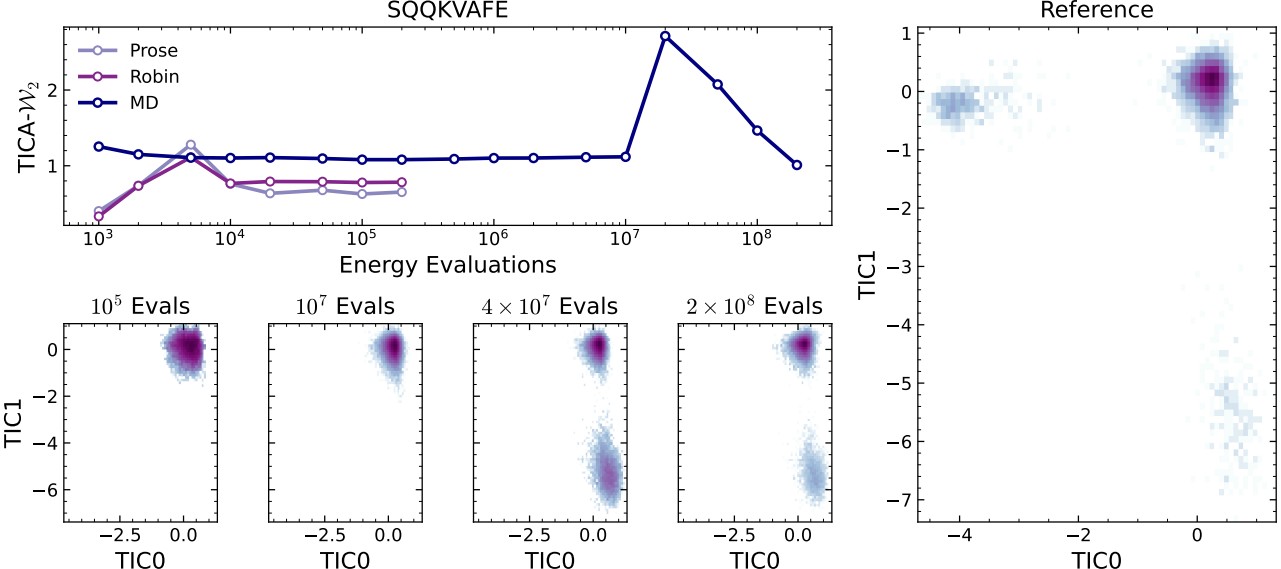

*Figure A.15.* How the TICA-$\mathcal{W}_2$ varies as a function of energy evaluations across MD, Prose, and ROBIN. In addition, in the bottom row we see how the MD simulation slowly discovers modes as more energy evaluations are performed; in this process, it often searches in incorrect regions, amplifying the TICA-$\mathcal{W}_2$, and then recovering from it with additional samples.

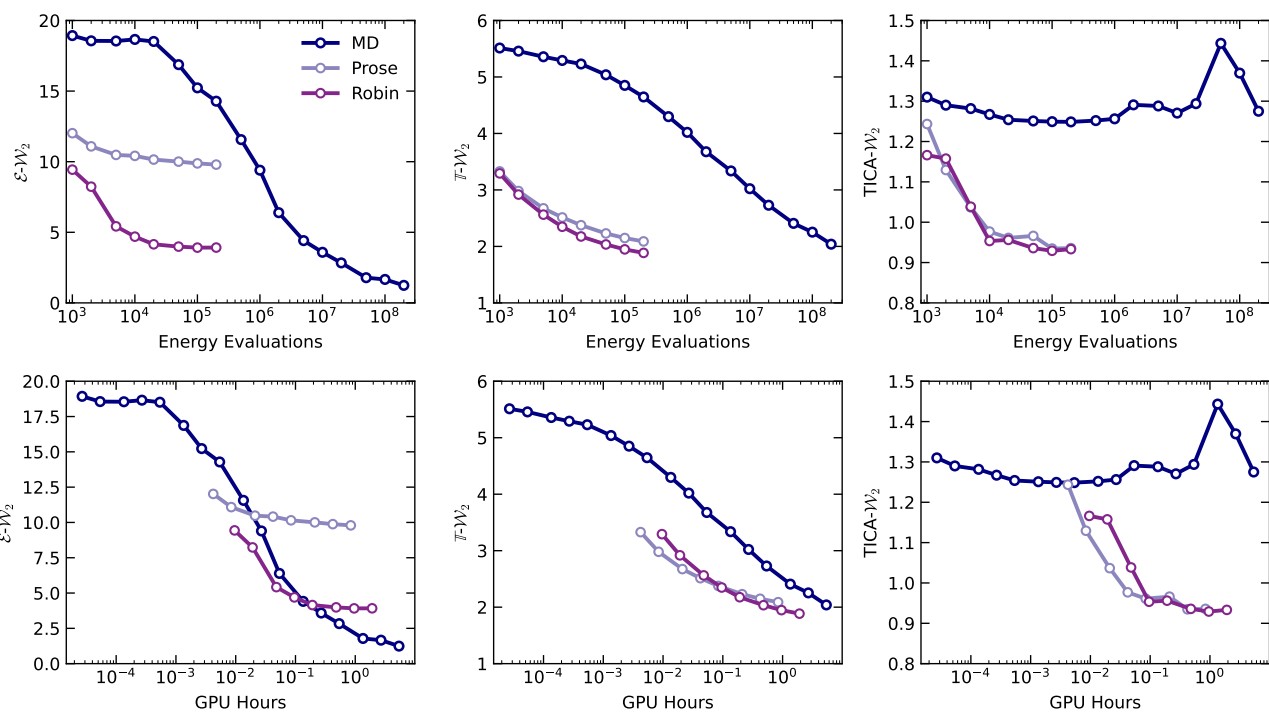

*Figure A.16.* $\mathcal{E}$-$\mathcal{W}_2$, $\mathbb{T}$-$\mathcal{W}_2$, and TICA-$\mathcal{W}_2$ against the number of energy evaluations and GPU hours for MD, Prose, and ROBIN.

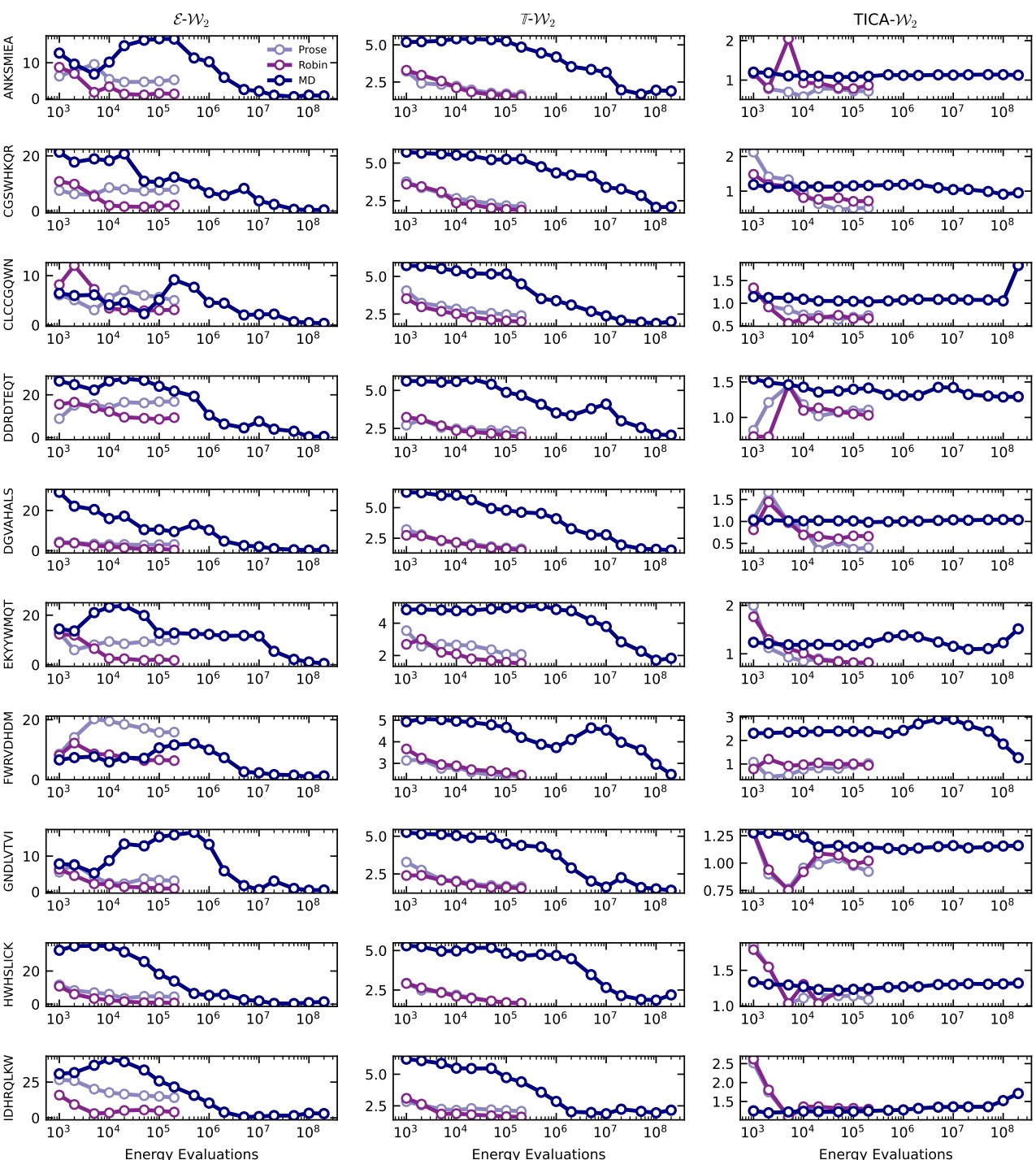

*Figure A.17.* $\mathcal{E}\text{-}\mathcal{W}_2$, $\mathbb{T}\text{-}\mathcal{W}_2$, TICA-$\mathcal{W}_2$ per peptide vs. Energy Evaluations.

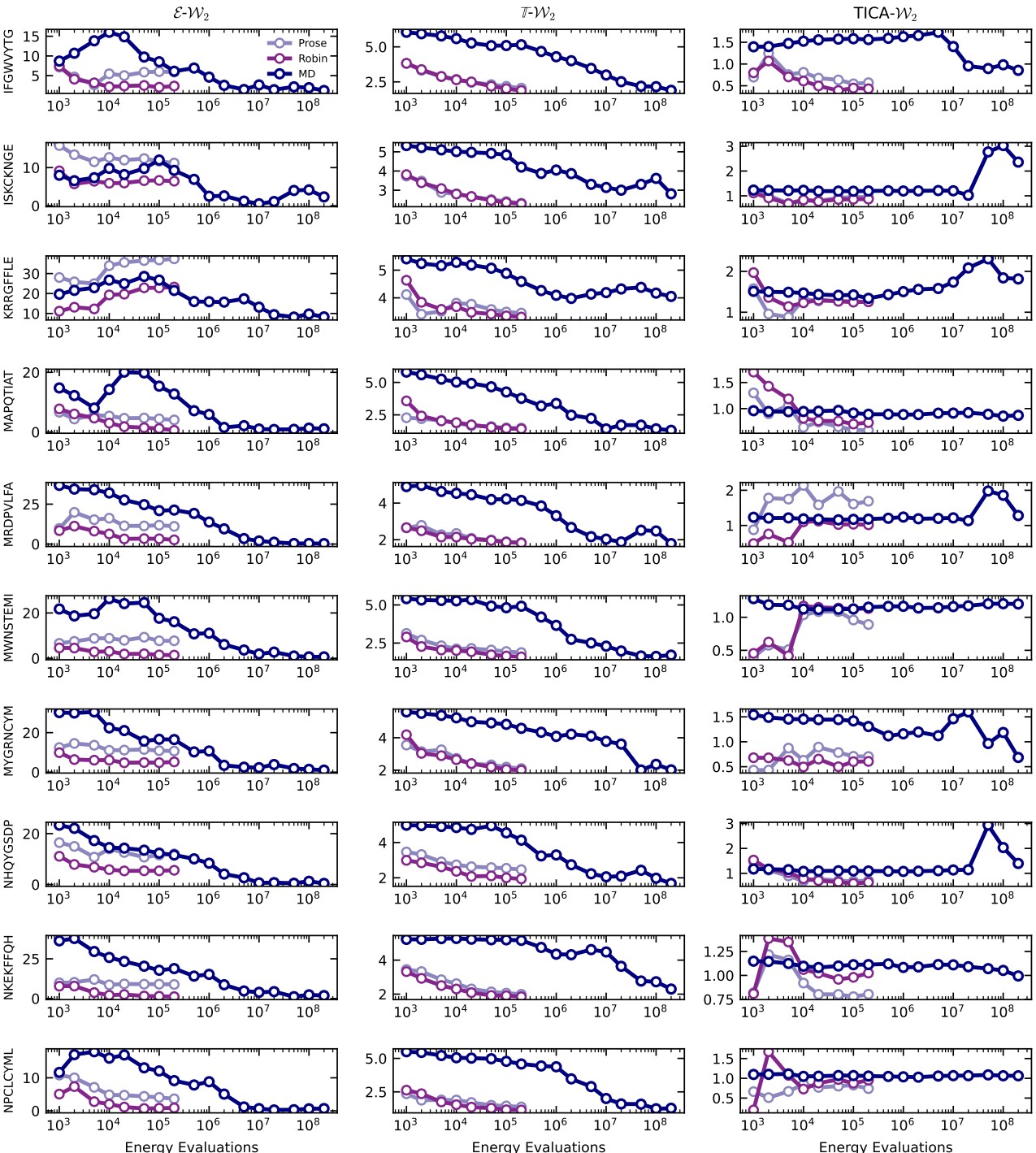

*Figure A.18.* $\mathcal{E}\text{-}\mathcal{W}_2$, $\mathbb{T}\text{-}\mathcal{W}_2$, TICA-$\mathcal{W}_2$ per peptide vs. Energy Evaluations.

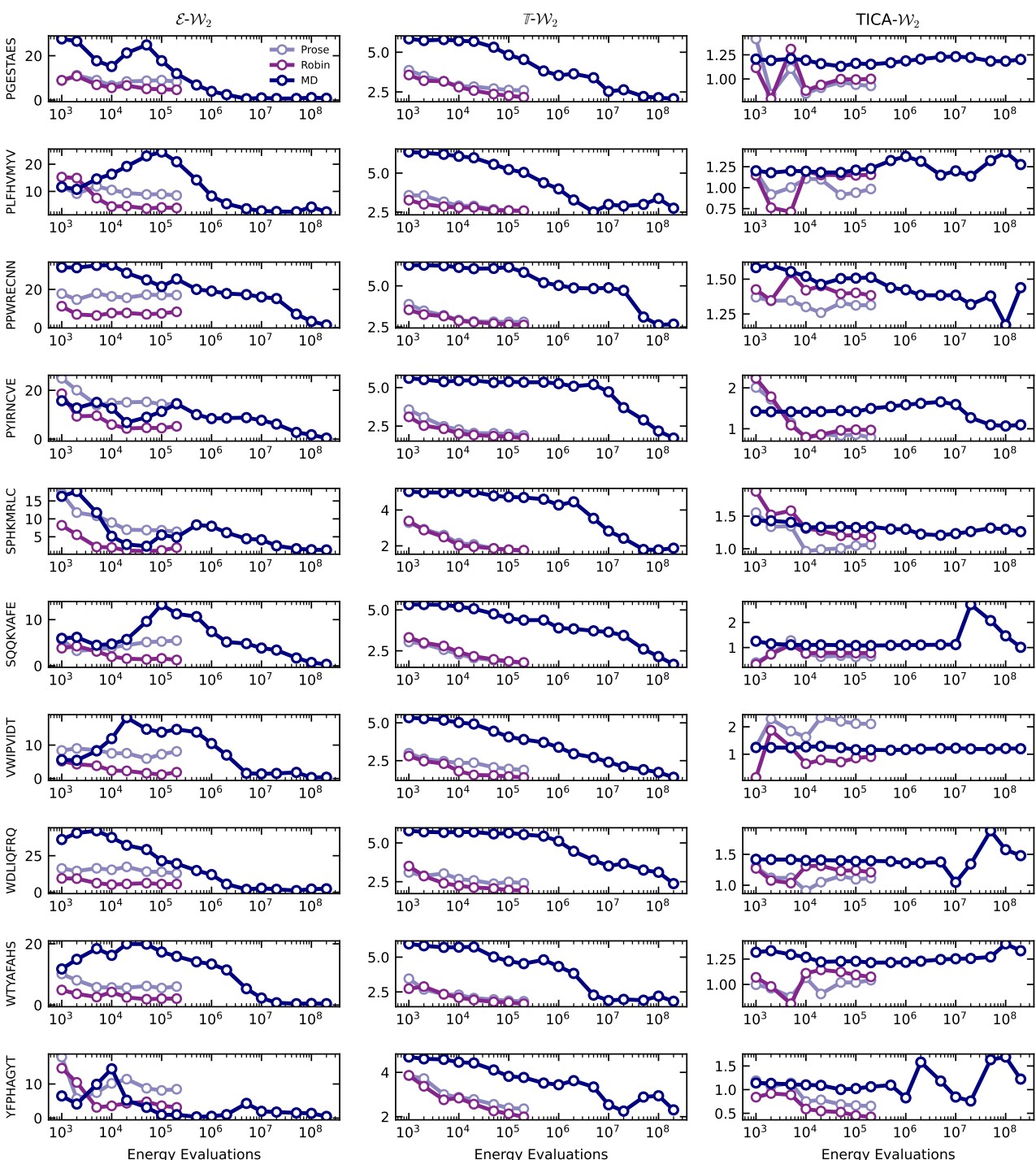

*Figure A.19.* $\mathcal{E}\text{-}\mathcal{W}_2$, $\mathbb{T}\text{-}\mathcal{W}_2$, TICA-$\mathcal{W}_2$ per peptide vs. Energy Evaluations.

# E. Experimental Configurations

## E.1. Architecture

The architecture choice employed follows a standard but performant recipe of Transformer-based building blocks. In Figure A.20, we capture the model specifications within a Transformer Block, which models the conditional distribution $p_\theta(x_j|x_{<j})$ and is composed of causal self-attention, RMSNorm (Zhang & Sennrich, 2019), and SwiGLU activations (Shazeer, 2020). The two main differences between the set of single peptide experiments and the transferable setting were: (1) the scale of the model; and (2) the conditioning, which we cover in detail below. Unique to the molecular setting, we include an additional source of conditioning information through embeddings of the atom type and residue types that are injected into the main transformer block. We discuss the details behind all the enhancements and best practices below.

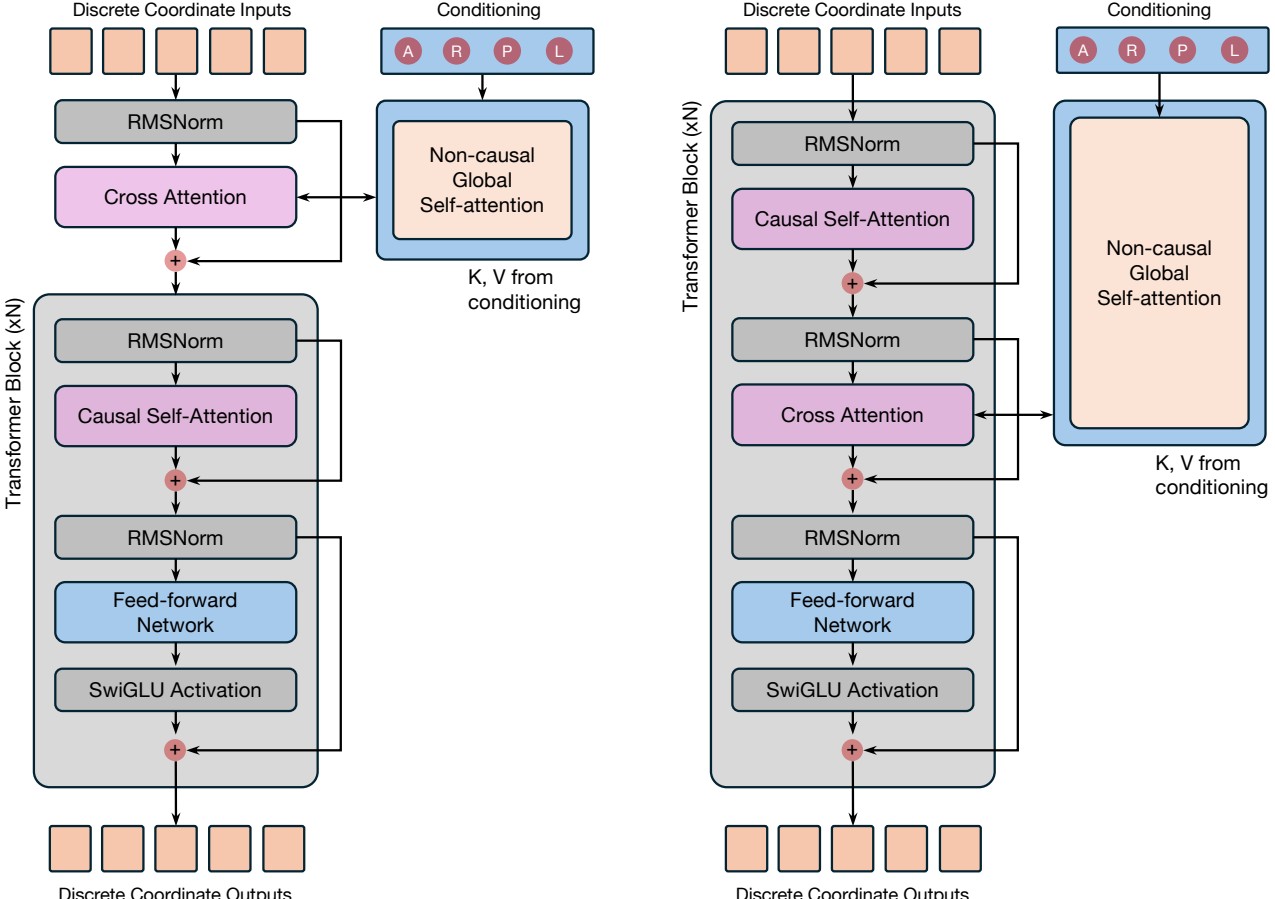

*Figure A.20.* The transformer-based architecture variants that were considered. **Left**: The decoder-only like architecture that takes the conditioning information in at the initial generation step; **Right**: The encoder-decoder like architecture that repeatedly has a cross attention block that interacts with every transformer layer.

## E.2. Conditioning

In the transferable setting, we explore various conditioning strategies. In line with Tan et al. (2025b), the conditioning information considers: (1) atom type, $A$; (2) residue type, $R$; (3) residue position, $P$; and (4) sequence length, $L$. To feed this conditioning information into the model, we consider two approaches that capture the ethos of encoder-decoder and decoder-only architectures commonly employed in modern LLMs.

More specifically, in the decoder-only variant (see left schematic of Figure A.20), we pass the conditioning information through a separate transformer model with non-causal masking (token by token predictions should have access to global conditioning information). The representations obtained from the transformer are subsequently injected directly into the first layer of the larger transformer. For language, most conditional information is passed in at the beginning of the model as

context, with additional passing at each layer excluded.

For encoder-decoder models—commonly adopted for settings like machine translation—where global conditioning information is useful for model predictions, two separate transformers are used for both encoding and decoding, with the interaction between them governed by a learned cross attention module. In a similar fashion to the decoder-only variant, we use a non-causal transformer to obtain representations for the conditioning information, and then compute cross attention by varying the query vectors of our larger model for fixed key and value vectors from our non-causal transformer. This approach allows us to more explicitly distinguish between information types (coordinate-type information being processed by the main transformer vs. residue-level information being processed by the non-causal transformer).

### E.3. Model Sizes

**ARBG and ROBIN**. For all single peptide and transferable generation experiments, we concluded upon the model configurations reported in Table A.5. For ease of contrast, we include the configurations used for competing models: SBG and Prose in Table A.4. In addition, although the model configurations appear identical between all alanine datasets for ARBG and ROBIN, the difference in parameter count can be attributed to the number of bins used. As stated in Table A.5, for alanine dipeptide and tri-alanine, we use 4096 bins, while for alanine tetrapeptide and larger, we use 8192 bins.

*Table A.4.* SBG and Prose configurations across molecular systems (Tan et al., 2025a;b).

| System | Layers / Block | Blocks | Channels | Parameters (M) |
|---|---|---|---|---|
| SBG (Alanine Dipeptide) | 4 | 4 | 256 | 13 |
| SBG (Tri-alanine) | 6 | 6 | 256 | 29 |
| SBG (Alanine Tetrapeptide) | 6 | 6 | 384 | 64 |
| SBG (Hexa-alanine) | 6 | 6 | 384 | 64 |
| SBG (Chignolin) | 8 | 8 | 384 | 114 |
| Prose (Transferable) | 8 | 8 | 384 | 285 |

**MoL/GMM-PixelCNN++ and GIVT**. For fair comparison, with ARBG we inherit the same de-quantization strategy and architectural blocks as ARBG when constructing the MoL/GMM-PixelCNN++ baselines. For all single peptide experiments, we set the bin count to $|B| = 2048$. For GIVT, as it is a fully continuous model in the vein of a true Mixture Density Network, there is no de-quantization needed. In all three baselines, the model includes an additional output projection head that outputs the parameters of the mixture distribution and has the shape:

$$\text{outputproj} = 3K + 2, \tag{25}$$

where $K$ is the number of mixtures, and for each mixture we output the means, scales, and logits over the mixture components. Lastly, we use two additional parameters as dependency coefficients to model the linear dependency coefficients for better modelling of correlated coordinates. In each case, the models are trained by computing the negative log likelihood under the mixture distribution. All remaining training settings are identical to the main model ARBG for single peptide systems.

### E.4. Training

**Optimizer, Learning Rate, and Scheduler**. Following the recent success of the Muon optimizer in accelerating LLM training (Jordan et al., 2024), we adopt it in our experiments. Muon is a momentum-based optimizer that applies Newton–

*Table A.5.* ARBG and ROBIN configurations for single system experiments and transferable sampling.

| System | Heads | Head Dim. | Layers | Channels | Expansion | Parameters (M) | Bins |
|---|---|---|---|---|---|---|---|
| ARBG (Alanine Dipeptide) | 8 | 32 | 8 | 256 | 4 | 7.4 | 4096 |
| ARBG (Tri-alanine) | 8 | 32 | 8 | 256 | 4 | 7.4 | 4096 |
| ARBG (Alanine Tetrapeptide) | 8 | 32 | 8 | 256 | 4 | 10.5 | 4096 |
| ARBG (Hexa-alanine) | 8 | 32 | 8 | 256 | 4 | 10.5 | 8192 |
| ARBG (Chignolin) | 8 | 64 | 12 | 512 | 4 | 39.9 | 8192 |
| ROBIN (Transferable) | 8 | 64 | 16 | 768 | 4 | 132.0 | 8192 |

Schulz orthogonalization to gradient updates. For weight matrices, it maintains an orthogonalized momentum buffer computed using five Newton–Schulz iterations, with Nesterov momentum ($\mu = 0.95$) and a learning rate of 0.02. For one-dimensional parameters, such as biases and normalization layers, the optimizer falls back to AdamW (Loshchilov & Hutter, 2019) with a learning rate of 0.002 and $(\beta_1, \beta_2) = (0.9, 0.999)$. We apply decoupled weight decay of 0.01 to all parameters and combine the optimizer with a cosine learning rate schedule with a warm-up phase covering 5% of the training iterations.

**Flash Attention**. To improve training efficiency and reduce memory consumption, we employ FlashAttention for all models (Dao, 2024). FlashAttention computes the attention operation using fused kernels that significantly reduce the number of memory reads/writes by avoiding the explicit materialization of attention matrices. This substantially reduces memory overhead and improves throughput, enabling faster training and better hardware utilization without altering the underlying attention computation or model behaviour.

**Lower precision training and inference**. We tested training in both bf16 and float32. We found that for smaller models, bf16 performed equally well to float32 training. However, for larger models, and primarily ROBIN, we observed that bf16 training sometimes resulted in training instability. We therefore chose to train ROBIN using float32 precision. We also found that for inference, float32 inference was more consistent than bf16, which anecdotally had some numerical irregularities.

**Hardware**. Training and inference was performed across multiple heterogeneous clusters containing a variety of NVIDIA GPUs. We primarily utilized L40S and RTX6000Pro GPUs for inference and training of single-system ARBG models and H100/H200 GPUs for training of ROBIN.

### E.4.1. SINGLE-SYSTEM

For the Chignolin scaling plot in Figure 1, we removed the scheduler, fixed the learning rate to $3 \times 10^{-4}$, and set the batch size to 256 across all trained models to ensure a fair comparison.

### E.4.2. ROBIN TRAINING

We train ROBIN using the same number of training steps as Prose, with a comparable batch size of 448 (7/8 of the Prose batch size 512). We use a learning rate $5 \times 10^{-4}$, with a cosine annealing schedule without weight decay.

## E.5. Inference

### E.5.1. AUTOREGRESSIVE TWISTED SEQUENTIAL MONTE CARLO

In this section, we detail our usage of SMC and the details on how it is applied in our peptide setting. In the ideal setting with a terminal reward $\mu_{\text{target}}(x)$, we would directly have access to the optimal intermediate density i.e.

$$\eta_j^\star \propto p_\theta(x_{\leq j})\psi_j^\star(x_{\leq j}). \tag{26}$$

with optimal twist functions:

$$\psi_j^\star(x_{\leq j}) \propto \sum_{x_{>j}} p_\theta(x_{>j}|x_{\leq j})\mu_{\text{target}}(x) \tag{27}$$

However, these optimal twist functions are, in general, difficult to obtain. While many works attempt to learn them using a variety of objective,s including soft $Q$-learning (Mudgal et al., 2024), noise contrastive estimation (Lawson et al., 2022), and classification (Yang & Klein, 2021), we already have a reasonable twist function using pre-defined energy functions. We approximate the twist function by the relative likelihood of a sample under the target energy function and our model. To encourage samples that are lower energy during our autoregressive generation.

We use the intermediate signals of a peptide-based energy function (either Amber ff99SBildn or Amber 14, depending on the system) (Tan et al., 2025a). However, these peptide-based energy functions only function correctly on complete peptides. In the case of a partial peptide (for instance, the subset of atoms PPW in the peptide PPWRECNN), these atoms are not able to be processed by the energy function because they do not form a complete peptide, which has a C-terminus oxygen cap, often denoted as OXT, the terminal oxygen.

We are therefore not able to efficiently evaluate the partial energy of a subsequence like PPW. Instead, we generate one more atom, the nitrogen atom of the next residue, which can tell us a reasonable direction for the oxygen atom to go. Our

procedure is then as follows:

1. Generate dimensions until we have generated a full residue plus the nitrogen atom of the next residue

2. Find the direction of the nitrogen atom relative to the carbon atom its attached to

3. Replace this nitrogen atom with an oxygen atom in the same direction off of the carbon atom, but at the optimal distance for carbon-oxygen bonds at $0.125$ nanometers.

4. Evaluate the energy of this subset of the peptide using the relevant Amber energy function.

This procedure allows us to calculate the intermediate twist functions, which then allows for resampling to improve the distribution during autoregressive sampling.

We perform SMC over the entire batch of samples for the best performance. This is much larger than can fit on a single GPU. Therefore, we need to operate in batches. We perform batched generation with KV caching per residue generated for minimal slow down. This has a couple of advantages over other generation procedures. First, we only need to hold a single gpu batch worth of Keys and Values at once. Second, we only need to regenerate the cache every residue. This means for a large batch of B samples, we need to regenerate the cache at most $(L-1)B$ times, where $L$ is the length of the peptide. This minimizes the additional overhead of SMC to a few more model evaluations, which is less than a 10% overhead.

### E.5.2. KV-CACHING

KV-caching is a technique used during inference that stores the attention keys and values from previous tokens during decoding to prevent recomputing them at every step (Pope et al., 2023). By caching these tensors, the model avoids redundant attention computations over the entire prefix, reducing per-token complexity and lowering latency at inference time. Consequently, we adopt it to reduce inference time cost when generating samples from the equilibrium distribution.

### E.6. Table and Figure Specific Details

**Table 1**. ECNF++, RegFlow, SBG, FALCON-A, and FALCON results are taken from Tan et al. (2025a) and Rehman et al. (2026a). SBG here is SBG with Sequential Monte–Carlo sampling instead of self-normalized importance sampling (SNIS), as this performed slightly better. All other methods utilize SNIS. ECNF++ dashes represent models that were not run on that system due to scaling concerns.

