# OpenReview forum: "Autoregressive Boltzmann Generators"
_ICML.cc/2026/Conference — ICML 2026 spotlight_

### Official Review · Reviewer_xhTd · 2026-03-02

**Soundness:** 3
**Presentation:** 3
**Significance:** 3
**Originality:** 3
**Overall Recommendation:** 5
**Confidence:** 4

**Summary:**

The authors propose autoregressive Boltzmann generators (ARBG), which combine cheap likelihoods with high expressivity. To model the conditional distribution in the autoregressive architecture, the authors suggest two main approaches: Discretized mixtures and uniform binning. Their approach circumvents the topology constraints present in flows due to them being homeomorphisms. The authors test the proposed architecture on multiple single-system benchmarks, where it outperforms all previous baselines. Furthermore, they test it in the transferable setting, where it also performs favorably, especially for the larger systems.

**Compliance With Llm Reviewing Policy:**

Affirmed.

**Final Justification:**

The authors addressed my concerns regarding the evaluation. The evaluation is now convincing, which is why I updated my score.

**Key Questions For Authors:**

7. Regarding Figure A.6. and the following statement: “We demonstrate that training models for too long leads to overfitting on the training data, which despite improving resampled metrics, yields undesirable behaviour.”: From the provided plots, it seems that not only does ESS not deteriorate despite the mode collapse, but also the torus metric T-W2 does not capture the mode collapse at all. Doesn’t this mean that the metrics chosen by the authors do not adequately capture mode collapse at all?
8. How is rotation- and translation-invariance of the target distribution handled when modeling it with the proposed autoregressive approach?
9. The inference temperature has been adjusted using a sweep. Has this been performed only for ROBIN, or also for the non-transferable models? Has this also been performed for the baselines, such as Prose? While the temperature adjustment is interesting, it potentially makes direct comparison difficult if it was not also applied to the baselines. Since T=1 seems to be close to optimal (at least this is what it looks like from the provided figures), this is not a large problem, but it should, at the very least, be stated in the text.
10. Have the authors considered a true mixture density modeling of the conditional probability, without binning? I believe this would be an interesting baseline. Can the authors explain why the binning of the mixtures was included in the first place?

**Limitations:**

yes

**Strengths And Weaknesses:**

**Strengths**

- At the moment, BGs based on discrete flows are the only ones that allow cheap exact likelihood evaluation. To the best of my knowledge, the proposed autoregressive architecture is the first, next to discrete flows, that also allows cheap reweighting. I believe this to be a very valuable and novel idea.
- The inclusion of partial energy evaluations as twist functions for SMC is very interesting, even though the efficiency gain is currently marginal. I have not seen such an approach before in the BG literature, and I am interested to see further developments in this direction.
- Empirical results in the non-transferable setting appear very strong (Table 1). However, I have some concerns about the chosen evaluation, see below.
- The transferable model “ROBIN” outperforms the previous SOTA in most metrics, especially for the larger systems. The provided inference-time scaling plots appear convincing.
- The provided ablations on bin resolution and sampling temperature are interesting and appreciated.
- The authors compare against a wide range of baseline methods.

**Weaknesses**

1. The authors' evaluation is limited, and I think many improvements should be made to enable a fair comparison with prior work and to allow better judgment of the proposed method.
   1. All main metrics (energy-based, torus-based, and tica-based) analyse low-dimensional marginals. I fear that even when overfitting to the training dataset, these metrics can all still look excellent, especially in the single-system experiments. ESS is unfortunately also not reported, as discussed below. An easy way to reassure that the model is not overfitting would be the inclusion of the negative log-likelihood on the train and test datasets as a function of training time. The proposed architecture should easily allow calculation of this metric.
   2. I find the complete exclusion of ESS to be unjustified. While the discussion by the authors correctly notes that ESS can be misleading under mode collapse and may be sensitive to outliers, these limitations are well known in the importance sampling literature and do not warrant omitting one of the primary diagnostics of sampling efficiency. ESS is a standard and widely reported metric for importance sampling methods. Excluding it substantially weakens the empirical evaluation and makes comparison to prior work difficult. Unless the authors can provide a principled argument for why ESS is fundamentally inappropriate in their setting (rather than merely imperfect), it should be reported.
   3. All metrics are reported using either SNIS or SMC. I propose to also include non-resampled raw proposal metrics, too. This would also quantitatively confirm statements such as the following one by the authors: “We attribute this to the high quality of the base ROBIN proposal. Since the model has learned the Boltzmann distribution sufficiently well, the SMC correction yields diminishing returns.”
   4. While TICA-W2 metrics are reported in some of the figures, I suggest including this metric also in the presented results tables for all systems, potentially in the appendix.
   5. Regarding the comparison of SNIS, SMC, and twisted SMC:
      1. Why was SMC for 2e5 evals omitted in Table 2?
      2. Why is there no comparison of twisted SMC vs standard SMC with ROBIN in Table 2? The authors state a roughly 3% efficiency gain of twisted SMC, but the impact on final metrics is not discussed. Do final metrics suffer?
      3. The limited increase in efficiency (3%) of twisted SMC should be stated and shortly discussed in the main text, not only in the appendix.
2. I think the statements regarding the limitations of flows (both continuous and discrete), with focus on them being homeomorphisms that preserve the topological structure, are too strong and lack evidence. Multiple recent works had great success using flow-based approaches as BGs, both using continuous-time and discrete-time flows. While flow-based architectures cannot, as stated correctly by the authors, place zero probability mass between modes, it is not evident that this is the reason for instabilities of such models during training of flow-based BGs. In practice, the modes in molecular systems are indeed not entirely disconnected (which is why transitions are possible after a long simulation time). I am not convinced that the fact that flows need to “stretch” space and place very low probability mass between modes is really a problem, and I am not aware of any theoretical or empirical analyses in this direction.
   1. The authors seem to suggest that this “problem” of having to stretch space is only relevant for discrete NFs, but not for continuous NFs \- there, as stated by the authors, the problem is only present during inference? I cannot really follow this line of reasoning, and at the very least, some references or additional explanations are missing here. Aren’t all NFs (continuous and discrete) homeomorphisms? Why is there a difference?
   2. I am also confused by the “expressivity bottleneck” stated for discrete NFs. In the transferable setting, the state-of-the-art baseline (Prose) that the authors compare against is based on a discrete NF (TarFlow). This seems contradictory.
   3. While the empirical results demonstrate strong performance gains of ARBG over flow-based baselines, the paper does not provide direct evidence that topology preservation is the primary bottleneck in those methods. It remains plausible that optimization challenges, architectural choices, or training objectives account for the observed differences.

**Other comments**

3. In paragraph “Conditionals as Discretized Mixtures”: $\\tilde{x}_j$ is first simply the clipped version of the continuous variable, and then later seems to be defined as the center of the bin. I think the notation can be improved; I find it partially confusing. Also, as far as I can tell, $\\mu\_k$ is not defined.
4. BGs based on TarFlow also allow the use of “standard LLM techniques”. Since this is stated several times in the manuscript as a key advantage of ARBG, one should mention that this can also be leveraged for flow-based BGs (TarFlow). Also, when the temperature scaling and KV-caching are discussed, I recommend mentioning that this was also already proposed for TarFlow before.
5. Have the authors trained TimeWarp and BioEmu themselves on the same datasets as the other models? Or are these models pretrained?
6. I don't see a reproducibility statement in the current manuscript. Knowing that the source code and data to reproduce the shown experiments will be made publicly available in the future would improve the contribution to the field.

---

> ### Author Rebuttal · Authors · 2026-03-31
>
> We thank the reviewer for their rigorous review. We are thrilled they found ArBG "very valuable" and "novel", twisted SMC "interesting", and our single peptide results "very strong", outperforming "previous SOTA in most metrics". Rebuttal PDF: https://anonymous.4open.science/r/arbg-7892/Rebuttal.pdf
>
> ## Limited evaluation
>
> We thank the reviewer for the detailed suggestions regarding additional experimental evaluations. We now include (1) test-set likelihoods (NLL and bits-per-dimension BPD), (2) ESS, (3) non-resampled metrics (Prop. Torus W2 & Prop. TICA W2) metrics in Table R3. We find Robin is the best on 8/10 of the new metrics across 4AA and 8AA systems, with the most noticeable improvements on 8AA and 4AA likelihood and Torus metrics. We believe these results significantly strengthen our claims especially around the quality of the base ROBIN proposal.
>
> **NLL curves:** Extracting full training/test curves over time retroactively requires prohibitively expensive reruns outside our compute constraints for the rebuttal. However, our final test NLLs demonstrate that ArBG does not overfit in the transferable setting.
>
> **TICA-W2:** We agree adding TICA-W2 to the single-system results would improve the paper. We will include them in future drafts as they require significant resources to rerun within the rebuttal period.
>
> **Twisted SMC:** SMC for 2e5 evals and Twisted SMC for ROBIN were omitted from Table 2 due to compute and time constraints during original submission. Our main goal was to showcase this inference-time avenue for ArBGs. While theoretically advantageous, our current approach yields small speedups. We will update the text to reflect this.
>
> ## Limitations of Flows
>
> We value the reviewers' constructive feedback on our initial claims on flows, which might have been too strong, and as noted by the reviewer, there may be other explanations for the performance gains of ArBG. We will revise the text to frame these points as hypotheses about scaling bottlenecks rather than proven causes of current instabilities.
> As the reviewer correctly points out, SOTA BGs are flow-based, and consequently, our main hypothesis is more forward-looking: identifying promising approaches for future scaling to even larger systems. In this regard, it is wholly unclear whether discrete NFs are the answer, as in addition to the architectural restrictions, they struggle with modelling non-smooth energy landscapes with varying topologies [Cornish et al. 2020, Dupont et al. 2019], while continuous flows have prohibitively expensive likelihood computation beyond our current scale.
>
> We clarify that while Prose was the state-of-the-art BG, this is due to efficient likelihood requirements, not based on expressivity. Prose is still far below the expressivity of SOTA protein models such as Alphafold3 and comparable continuous NFs according to our own experiments. The expressivity bottleneck is clear in images and video models which are based almost exclusively on continuous NFs. Our intention was to motivate diffeomorphism-free approaches to BGs, not to claim that we have direct evidence that topology preservation as the primary bottleneck. We understand how this was not clear and we will update the text accordingly.
>
> ## Comments
>
>
> * 3 and 4: We thank the reviewer and will update the manuscript accordingly.
> * 5: Following Prose, we use pretrained weights of TimeWarp and BioEmu.
> * 6: Source code to reproduce the shown experiments will be made publically available under a permissive MIT license.
> ## Questions
> * 7: We highlight that A.6 is an artefact of overfitting on ALDP, where the dropped mode is much smaller in weight than the main mode. This makes ESS particularly insensitive to detecting this phenomenon. The other metrics in this case are also slightly insensitive, as the mode itself is quite small and does not materially reflect in the metrics. In other systems, the metrics perform as desired, as there is more equal weightage across modes.
> * 8: In line with current literature of all-atom modelling, we softly enforce SE(3) symmetry via data augmentation and mean centering, which we found empirically sufficient for our autoregressive objective.
> * 9:  Temperature sweeps were only performed for ROBIN. We have now included $T=1$ in the rebuttal link in Table R2 and R5. As noted, this is not a large problem as the best temperatures were not that far from $T=1$.
> * 10: Mixture density modeling. We thank the reviewer for their insightful suggestion, which we tried. However, this performed measurably worse than our GMM and MoL results. We believe that this is due to the common failure mode of MDNs, where the mixture components often collapse to a small subset of the mixture components.
> ## Concluding remarks
> We thank the reviewer again for their time and effort during this rebuttal process. We hope our answers here have aided in clarifying all the great questions raised by the reviewer. We are also available to engage further.

---

> > ### Author Rebuttal · Reviewer_xhTd · 2026-04-01
> >
> > I thank the authors for their hard work during the rebuttal and appreciate all additional evaluations and answers to my questions. I am now convinced by the evaluation and am thus raising my score to “Accept”. I suggest adding the extended evaluation to the manuscript, as it gives a more complete picture and considerably strengthens the contribution.
> >
> > That said, I have some final comments:
> >
> > - Regarding the expressivity bottleneck of discrete normalizing flows: “We clarify that while Prose was the state-of-the-art BG, this is due to efficient likelihood requirements, not based on expressivity. Prose is still far below the expressivity of SOTA protein models such as Alphafold3 and comparable continuous NFs according to our own experiments.”
> >   - If I see this correctly from the Prose paper, Prose outperformed ECNF++ given the same budget of target evaluations (independent of the likelihood eval cost). Is the implication here that the CNF architecture needs to be chosen small enough to still make likelihoods viable, and this is why a discrete normalizing flow wins in this case?
> >   - In the end, this is just a matter of framing the discussion correctly. I suggest that the authors only include claims if they can be clearly supported by experiments in the current manuscript or related work. Otherwise, as the authors already suggested, they should be phrased as hypotheses or entirely omitted.
> > - “Following Prose, we use pretrained weights of TimeWarp and BioEmu.”
> >   - Does this mean they were trained on different datasets? If this is the case, I find direct comparisons hard to justify. I suggest either highlighting that these architectures were trained on different data sources in the main results table, or moving these results to the appendix, with a clear note to be cautious about direct comparisons.
> > - “In line with current literature of all-atom modelling, we softly enforce SE(3) symmetry via data augmentation and mean centering, which we found empirically sufficient for our autoregressive objective.”
> >   - I might have missed it, but I believe this was not discussed in the original submission and should thus be added.
> > - “Temperature sweeps were only performed for ROBIN.”
> >   - I thus suggest reporting only T=1 in the main table of the manuscript for a fair comparison of the architectures. The temperature sweeps are interesting, but should be presented as an additional ablation in the appendix.
> > - “Mixture density modeling. We thank the reviewer for their insightful suggestion, which we tried. However, this performed measurably worse than our GMM and MoL results.”
> >   - I’m not sure if this was part of the original submission, but I suggest at least mentioning that this was tried and that it performed measurably worse. The binning should be somehow motivated; otherwise, readers might be confused by this choice.
> > - Regarding twisted SMC: I still suggest adding some experiments that directly show the impact on the final metrics. Currently, only speedup is measured, but it is unclear if the metrics are affected. Since I believe this to be a very interesting idea, even though it is only on the side of the main contribution, such analysis could be very valuable for future developments in similar directions.

---

> > > ### Author Response · Authors · 2026-04-01
> > >
> > > We thank the reviewer very much for their time and thoughtful engagement during the rebuttal period, and for raising the score to "Accept." We are delighted that our additional evaluations and responses have fully resolved your concerns. The constructive feedback has undeniably strengthened the empirical evaluation and the clarity of our paper. We now address the new comments.
> > >
> > > We completely agree with the additional comments and will incorporate these changes into the next version of the manuscript.
> > >
> > > ## Framing of Discrete NFs and Expressivity
> > > The reviewer makes an insightful point regarding the nuances of the Prose vs. ECNF++ comparison. We agree that the success of discrete NFs in that context is largely due to the restrictions placed on CNFs to keep likelihoods computationally viable. For instance, we can construct a crude estimate via a back-of-the-envelope calculation on our previously trained, similarly sized flow matching model on transferable (322M parameters). This model takes approximately 9.3 seconds per sample on an L40S GPU on the smallest test system (AA). Prose is approximately 0.004 seconds per sample on the same system (~2000x faster). For larger systems, this is exacerbated. A very rough estimate puts this at >100,000 L40 GPU hours to evaluate a CNF-based BG of approximately equivalent size on the transferable test set for SNIS over 200k points.
> > >
> > > ## Pretrained Baselines (TimeWarp and BioEmu)
> > > We agree that comparing models trained on different datasets requires clear caveats. We will add a prominent footnote/asterisk to the transferable results table (Table 2) explicitly stating that TimeWarp and BioEmu utilize pretrained weights from different data distributions. We will also add a dedicated note in the text urging caution when making direct comparisons with these specific baselines.
> > >
> > > ## SE(3) Symmetry Enforcement
> > > Thank you for catching this omission. We will add a paragraph to the experimental configurations section (Appendix E) explicitly detailing how we handle rotation- and translation-invariance via data augmentation and mean-centering.
> > >
> > > ## Mixture Density Modeling Motivation
> > > Thank you for the suggestion. We will add a note in the text introducing the discretized mixtures. We will explicitly mention that standard continuous mixture density modeling (MDNs) was evaluated initially but performed measurably worse due to severe mode collapse, which serves as the primary motivation for our binning approach.
> > >
> > > ## Twisted SMC Final Metrics
> > > We completely agree that evaluating the impact of twisted SMC on the final metrics is valuable for future research. We will add a larger empirical analysis of performance with respect to the final metrics in the appendix.
> > >
> > > We thank the reviewer once again for engaging in the review process and for the constructive, detailed, and impactful feedback on our work. We value the reviewers' comments that have helped improve the quality of our work.

---

### Official Review · Reviewer_NPzH · 2026-03-10

**Soundness:** 4
**Presentation:** 4
**Significance:** 4
**Originality:** 4
**Overall Recommendation:** 5
**Confidence:** 3

**Summary:**

This paper proposes a novel approach for equilibrium sampling of molecular systems using autoregressive models instead of normalizing flows. The authors argue that flow-based Boltzmann Generators suffer from topological constraints and numerical instability when handling multimodal distributions with separated metastable states. The paper addresses this by discretizing molecular coordinates into bins and predicting them autoregressively using transformer architectures. They also introduce ROBIN, a 132M-parameter transferable model, claiming ~60% improvement in zero-shot energy error. The paper demonstrates strong results on Chignolin and introduces Autoregressive Twisted SMC for early rejection of physically implausible substructures during generation.

**Compliance With Llm Reviewing Policy:**

Affirmed.

**Final Justification:**

This paper proposes a novel approach for equilibrium sampling of molecular systems using autoregressive models instead of normalizing flows. The paper is nicely written and has extensive experimental evaluation. During the rebuttal, the authors answered my concerns positively. I am in favour of acceptance.

**Key Questions For Authors:**

- **Scaling analysis for larger systems**: Can you provide scaling analysis for systems >10 residues? Can ROBIN even accommodate 8192 bins for a 50-residue protein? Empirical power laws relating protein length, accuracy, and computational cost would significantly strengthen the paper, in my opinion.

- **Wall-clock inference time**: For practitioners, actual compute time matters as much as energy evaluations. From my understanding, ROBIN requires 3× model evaluations since it operates on n×3 dimensions rather than atoms. Therefore, ROBIN's advantage exists only in energy evaluations, not computational cost. The main text downplays this critical limitation and only briefly mentions it in section D.2. Can you make the statement clearer to the reader?

- **Oracle performance bounds**: Can you provide Oracle results to quantify the performance lost to binning? It would be valuable to compare against the Proposition 1 upper bound.

- **Temperature selection mechanism**: Is the optimal temperature predicted internally by the model, or is it set manually through sweeps for each new system? Clarifying this would help assess the "zero-shot" transferability claim.

- **ESS metric exclusion**: I do not fully understand why ESS is excluded from the main benchmarks. Is the mode coverage claim robust despite this metric choice? I suggest including the ESS results in the appendix to allow readers to evaluate mode coverage independently.

**Limitations:**

yes

**Strengths And Weaknesses:**

### Strengths

- **Well-motivated and clearly written**: The paper is well-motivated and well-written, and it is easy to follow by introducing relevant previous works. The paper presents a novel approach using autoregressive models to overcome the topological constraints of normalizing flows.

- **Principled theoretical framework**: Proposition 1 rigorously bounds the approximation error from uniform binning in KL divergence terms, providing solid theoretical grounding for the discretization design.

- **Impressive experimental results**: ARBG dramatically outperforms baselines (Table 1), and Ramachandran plots (Figure 3) show near-perfect mode coverage. The authors also provide extensive ablations (bin resolution, temperature, de-quantization) and detailed per-peptide analysis that clearly demonstrate both the benefits and limitations of the methodology.

- **Novel architectural capability**: Autoregressive Twisted SMC enables intermediate rejection during generation, this capability is unavailable in flow-based methods.

- **Excellent scaling in energy evaluations**: ROBIN achieves competitive performance while requiring an order of magnitude fewer energy evaluations compared to Prose and three orders of magnitude fewer compared to MD.

### Weaknesses

- **Hard discretization limit**: Uniform binning fundamentally limits precision to bin width, as acknowledged in Proposition 1. I find the practical impact severe since the model cannot recover fine-grained structure regardless of training quality. While Figure A.2 shows nice convergence with finer discretization, larger systems with sharp energy profiles might be poorly approximated by this methodology (which the authors acknowledge in the limitations). I will not penalize the score for this weakness, as it is inherent to the discretization approach and was disclosed, but I would have liked to see some ablation studies (similar to Fig. A.2) for larger proteins.

- **Overstated scope of "state-of-the-art" claim**: The paper claims "state-of-the-art" for "molecular sampling" when restricted to ≤10 residues (Chignolin being the largest tested). Real proteins are 100–1000+ residues. There is no scaling analysis or preliminary results for larger systems. With 8192 bins and n×3 dimensions, I expect memory and inference costs would become prohibitive for realistic proteins. **Suggestion**: Include scaling analysis in the main paper and adjust claims (e.g., "SOTA for small peptides" rather than "molecular sampling" broadly).

---

> ### Author Rebuttal · Authors · 2026-03-31
>
> We greatly value the reviewer's time, energy, and positive appraisal of our work. We are particularly delighted that the reviewer found our paper to be “well-motivated and well-written” and that Prop 1 provides a “solid theoretical grounding” for ArBG. We are also grateful that the reviewer found ArBG to “dramatically outperform baselines” and that our transferable model Robin is scalable, as it requires “an order of magnitude fewer energy evaluations”. We now address the main questions raised in the review grouped by theme and post our rebuttal experimental results in the following link: https://anonymous.4open.science/r/arbg-7892/Rebuttal.pdf
>
> ## Ablation study for larger proteins
>
> > but I would have liked to see some ablation studies (similar to Fig. A.2) for larger proteins
>
> As suggested, we performed two ablation studies for a larger protein (Chignolin) which are located in the rebuttal link. We provide bounds on the energy and torus for different bin widths, and also demonstrate that increasing bin count improves performance through a dedicated experiment using a larger protein: Chignolin (see Figure R4). Further, the bin widths, detailed in Table R4 demonstrate that our bin widths are on the order of 0.3-0.5 pm, which enable recovery of fine-grained structures, and we provide a plot to demonstrate the upper bound in performance for our given bin widths in Figure R3.
>
> ### Claims and scaling to larger systems
> We thank the reviewer for this constructive feedback and the excellent suggestion to include a scaling analysis.
> First, we clarify that it was not our intention to claim SOTA performance for molecular sampling tasks broadly. Our claim was intended strictly in the context of outperforming the state-of-the-art Boltzmann Generators evaluated in our work (e.g., Prose / FALCON) on peptide systems. However, we agree that precision is critical, and we see how our original phrasing could be interpreted too broadly by readers and will carefully revise the manuscript (including the Abstract, Introduction, and Conclusion) to explicitly scope our claims.
> > Can you provide scaling analysis for systems >10 residues
> We agree that a study on larger proteins would be informative. Due to time and computational constraints of the rebuttal period, we were unable to run a full analysis for systems larger than Chignolin. We note that Chignolin is the largest system currently benchmarked by BGs, so any larger system would constitute introducing a new dataset as well.
> ### Inference cost
>
> In Appendix Figure A.16 we report all of our quantitative metrics against the number of energy evals and GPU hours for the transferable setting. We observe that Robin outperforms Prose on E-W2 with respect to both energy evaluations and walltime, and achieves comparable performance on the macrostructure metrics for the same compute costs. We hope this clarifies the reviewers' healthy concern regarding the computational efficiency of Robin comparison to Prose.
>
> > ROBIN requires 3× model evaluations since it operates on n×3 dimensions rather than atoms.
> We agree that Robin operates on dimensions rather than atoms requiring more 3x more model evaluations. However, there is a subtle point about the structure. Importantly, the prior state-of-the-art (Prose) also relies on autoregression, requiring 8 passes over different data permutations, whereas Robin uses only a single autoregressive pass with a fixed permutation to predict atom coordinates, making it comparatively more efficient per model evaluation (3x number of atoms vs. 8x number of atoms for Prose and Tarflow). We thank the reviewer for helping us clarify this to the reader.
> ### Oracle bounds
>
> This is a great suggestion. We have already included this in our manuscript in Figure A.2, which shows the energy distribution and torus metrics both quantitatively and qualitatively for various discretization levels.
>
> ### Temperature Selection
>
> Temperature is not set independently per system and is set through sweeps with a single temperature for all systems. Transferability to new systems is not dependent on temperature tuning (see additional results for $T=1$ in Table R5).
>
> ### ESS Metric
> We appreciate the reviewer’s concern regarding ESS. As discussed in C.2, ESS can be misleading in high dimensions as it is dominated by rare samples (e.g. clipping some percentage of large importance weights is necessary in all works). Nonetheless, given its historical use, we report ESS in Table R3. For the largest peptide systems—Chignolin (10AA), and 4AA and 8AA in the transferable setting—Robin outperforms baselines on Chignolin and 8AA, but trails Prose on 4AA, indicating competitive ESS performance on the two largest systems.
> ## Concluding remarks
>
> We thank the reviewer again for their time and positive feedback. We hope our rebuttal here adequately addresses the reviewer's key questions and enables them to continue endorsing our paper. We are also available to answer any further questions that arise.

---

> > ### Author Rebuttal · Reviewer_NPzH · 2026-04-02
> >
> > I thank the authors for answering my raised concerns. I am fully convinced by the experimental evaluation and in favour of accepting the paper.

---

> > > ### Author Response · Authors · 2026-04-06
> > >
> > > We thank the reviewer for their engagement in this discussion and for their support of our paper. We are glad that we could fully address all raised concerns. We will include all additional experiments in the final version of the manuscript.

---

### Official Review · Reviewer_5dtw · 2026-03-12

**Soundness:** 3
**Presentation:** 3
**Significance:** 3
**Originality:** 3
**Overall Recommendation:** 5
**Confidence:** 4

**Summary:**

The authors introduce an autoregressive Boltzmann generator that discretizes the continuous positional space of atoms into bins during training. Two discretization strategies are considered: discretized mixtures and uniform binning. Uniform binning was found to perform the best. The autoregressive model is then benchmarked against other Boltzmann generation methods on peptides ranging from dipeptides to decapeptides, with the autoregressive model (Robin) achieving competitive to state-of-the-art performance across all tasks.

**Compliance With Llm Reviewing Policy:**

Affirmed.

**Final Justification:**

I believe the paper is a pretty strong contribution to the field and the authors have addressed my concerns.

**Key Questions For Authors:**

I have the following questions/comments for the authors:

1) To calibrate the reader to the level of spatial resolution the authors should specify what the width of each bin is in terms of nm/Angstroms.
2) The authors compute an optimal sampling temperature for each system in the study. However, since this is tuned to best fit that particular system's test set, it does not speak to the generalizability of the method. The authors should also report results using a standard sampling temperature of 1 across all systems.
3) Some conclusions in Appendix D.3 are unclear. The authors claim that powerful generative models like Robin overfit to the positive $\varphi$ mode, yet the Ramachandran plot in Figure A.6 appears to show the opposite, with samples drawn from the negative mode. To clarify this, the authors should provide Ramachandran plots for both the proposal and reweighted distributions.
4) Also regarding the ADP experiment, which energy function is used to reweight samples generated samples? GFN-xTB or Amber? A mismatch in the location of the positive $\varphi$ mode between the two energy functions could cause that mode to vanish upon reweighting. Is the energy function consistent with the one used by the other methods in this comparison?

**Limitations:**

The authors have discussed the main limitations that come in mind, i.e, the imposed sequential nature of generation, and the need for better precision/higher bin counts in larger systems.

**Strengths And Weaknesses:**

Strengths: The paper is technically sound, has strong results and is presented well. The method of discretizing continuous spaces to get easier access to generated likelihoods is a potentially valuable contribution to the field. The method is evaluated comprehensively across multiple benchmarks, achieving competitive to state-of-the-art performance on all tested systems.

Weaknesses: Some of the conclusions specifically related to alanine dipeptide are unconvincing and requires further investigation.

---

> ### Author Rebuttal · Authors · 2026-03-31
>
> We are grateful to the reviewer for their time and effort in this review process. We greatly value that the reviewer found our paper to be “technically sound” with “strong results”, state-of-the-art results which are “evaluated comprehensively across multiple benchmarks”. We now address the main questions raised in the review and host rebuttal experiments in the following link: https://anonymous.4open.science/r/arbg-7892/Rebuttal.pdf
>
> ### Bin resolution in Angstroms
>
> We thank the reviewer for this comment and agree. We include the bin widths for all settings in Table R4 in the rebuttal link. For our transferable model, we use a bin width of 0.385 picometers (0.000385 nanometers or 0.00385 angstroms). The largest bin width is on AL6 at 0.499 picometers. We will add these details to the updated manuscript.
>
> ### Temperature T=1 results
>
> We appreciate the reviewer's concern regarding reporting results at temperature $T=1$. We report these in Table R2 and R5, and find ArBG is still able to outperform state-of-the-art across most systems. In the transferable setting, we observe that at $T=1$, Robin remains worse than Prose on 4AA and superior to Prose on 8AA in line with the results obtained at $T=0.95$.
>
> ## ALDP conclusions
>
> ### Robin overfits on ALDP
>
> To further support the claim of overfitting on ALDP we computed training and validation losses over training time in table R1. We find that validation loss shows classic signs of overfitting after epoch 239.  We include the proposal and reweighted Ramachandran plots in Figure R5 for clarity. We will include them in the appendix.
>
> To clarify Figure A.6 in our original appendix, the reviewer’s observation is correct that the reweighted Ramachandran drops the negative mode, which we do not see in the proposal Ramachandran plot. Overfitting causes the likelihood of generated samples in the positive mode to increase substantially, meaning that after reweighting these points disappear.
>
> ### ALDP Energy function mismatch
>
> We acknowledge the reviewer's comment about using a specific energy function in our ALDP setup. We highlight that we follow the same experimental setup as prior work, which uses the Amber energy function, which differs from the energy function used to generate the dataset. We agree that this may cause a discrepancy between the dataset and resampling, and perhaps this benchmark would be better served by evaluation with another energy function. We stress, is problematic as we note in our appendix and also one of the main reasons we wish to shift the community to use other single peptide datasets as a more standard benchmark. However, for transparency we still reported ALDP results in our appendix, despite these potential discrepancies with the energy function that reviewer correctly points out.
>
> Despite the potential discrepancy, we argue that mode droppage is unlikely to be due to a mismatch of energy functions. In particular, we argue that it is overfitting that leads to mode droppage, as we find in Fig A.6 of the appendix that the mode is captured early in training before it is lost in late training, which substantiates our argument that the energy function is not responsible for this behaviour.
>
> ### Concluding remarks
> We thank the reviewer again for their time and dedication to the rebuttal process. We believe our answers have clarified the main questions raised by the reviewer with regard to overfitting and our ALDP experiment in the appendix. If the reviewer is satisfied with our responses, we very politely encourage the reviewer to consider a fresher evaluation of our work with a potential score upgrade. We are also happy to answer any further questions.

---

> > ### Author Rebuttal · Reviewer_5dtw · 2026-04-03
> >
> > I thank the authors for the response and conducting more experiments on ALDP. I am now more convinced of the overfitting behaviour that the authors claim. As a control experiment, it would be nice to see how Robin performs on the original dataset that is **not** reweighted by the Von-Mises distribution - does that mode still disappear? I would request the authors to include such an experiment in the final version. I do think this paper is a very valuable contribution and I will raise my score to reflect this.

---

> > > ### Author Response · Authors · 2026-04-06
> > >
> > > We would like to thank the reviewer for their thorough review and insightful comments. In particular, we are glad that the reviewer is now more convinced with the additional experiments on alanine dipeptide. We agree that the suggested control experiment would be a valuable addition to this work and will include it in the final version as suggested. We thank the reviewer again for their engagement in this discussion which has led to a stronger, better supported paper.

---

### Official Review · Reviewer_6th3 · 2026-03-12

**Soundness:** 3
**Presentation:** 4
**Significance:** 3
**Originality:** 4
**Overall Recommendation:** 5
**Confidence:** 5

**Summary:**

The authors introduce a new type of Boltzmann generators that does not rely on (continuous) normalizing flows, but uses an LLM-inspired architecture. While flow-based approaches have had a huge success over the years, they often suffer from slow likelihood computation or are limited because they are diffeomorphisms. However, their autoregressive model allows for exact likelihoods in a single pass, and hence allows for scalability due to decomposition. To achieve this, they consider not a continuous space but discretize the space into bins spanning a categorical distribution and predicting one atom at a time. The final prediction is then a weighted sum of uniform distributions (one for each bin).

**Compliance With Llm Reviewing Policy:**

Affirmed.

**Final Justification:**

I think that the authors have fully addressed all of my concerns, especially regarding performance and their binning strategy. I am grateful that the authors commit to releasing their source code, increasing my confidence in their work and reproducibility.

I hope that the authors revise the parts in the text and add the additional/extended plots they provided. Especially, the additional analysis of alanine dipeptide overfitting.

Overall, I increase my score to "Accept".

**Key Questions For Authors:**

1. Why does Robin clearly outperform Prose in terms of the T-W2 metric in Fig. 1 and 16-A but not for the other metrics?
2. Fig. 5 shows that performance monotonically increases with bin count. However, I think there should be a point where the trend reverses because the number of bins will be too large / training data in each bin too sparse for the model to work well? How would you tune the number of bins? Did you observe any of this in your experiments?
3. For discretizing the space, you first need to know the maximum extent of the molecule. Does that put a limit to generalization to larger systems?
4. First of all, I would like to acknowledge that the authors did not try to hide this but referenced issues with alanine dipeptide in the main text. In Fig A.6 we can see that with continued training, the low-probability mode of alanine dipeptide disappears due to overfitting. You also show the training loss going down, how does the validation loss behave, does it also go down? Also, do you have any ideas how this can be prevented instead of early stopping? For instance, could a smaller / larger batch size fix this? I would be curious to know if the authors tried to alleviate these problems.
5. The main metric the authors were showing are either Wasserstein distances or free energy plots. However, the energy / torsion angles alone does not fully describe a molecule. Could the authors provide information / plots such as bond distances? I could imagine that the more bins are discretized, the loss accurate the bond histograms get.
6. It is not clear how the errors accumulate when placing many atoms one after another. In other words, what happens if an atom is misplaced early on during sampling? There is no way to fix this (besides rejecting the sample). Is there a way to measure this? I can imagine that this grows with atoms and can quickly become a bottleneck.

**Limitations:**

yes

**Strengths And Weaknesses:**

## Strengths
- The manuscript is excellently written, easy to follow, and is structured in such a way that it anticipates upcoming questions and answers them immediately.
- Conceptually, they introduce a novel idea to train a Boltzmann generator (BG) like an LLM which, as far as I am concerned, has not been explored before. Separation into bins allows for fast likelihood evaluation.
- They make use of energy-based resampling during the sampling process (“Twisted Monte Carlo”). They generate parts of the molecule and discard proposals which likely include sampling errors. While they say that it does not improve the final results by much, they say that this reduces steric clashes and saves on compute.
- They show state of the art results for transferable BGs, requiring less energy evaluations than their competitors.

## Weaknesses
- I think the main weakness is also their main novelty. Due to the introduction of the discrete binning of the true distribution, they will inevitably introduce approximation errors. While they show that smaller bins lead to less errors (as one would expect), I am not fully convinced that this is an optimal parameterization (see questions).
- During inference, each atom needs to be predicted individually and one after another. For larger systems, this can introduce a significant computational overhead. While we see that this scales rather well (e.g., LLMS can also write whole essays) one shot generation or joint modeling like in diffusion/flow matching could conceptually be beneficial.
- In Table 2, the improvements over PROSE only seem marginal.
- I do not see any reference to the code being published or it being uploaded in the supplementary materials.

---

> ### Author Rebuttal · Authors · 2026-03-31
>
> We are grateful to the reviewer for the time, care, and detailed feedback in evaluating our submission. We are pleased that the reviewer found our paper “excellently written”, and appreciated our “novel idea to train a BG like an LLM”, which led to “state-of-the-art results for transferable BGs”. We now address the main questions raised in the review and host rebuttal experiments in the following link: https://anonymous.4open.science/r/arbg-7892/Rebuttal.pdf
> ### Optimal Parameterization – Binning
> We agree that uniformly spaced bins over dimensions is almost certainly not an optimal parameterization. Furthermore, we believe it is an extremely exciting direction to explore other parameterizations beyond the simple ones considered. Moreover, the goal of this work is to show that, somewhat surprisingly, uniform discrete binning can be competitive, and often superior to, current flow-based approaches to BGs.
> With regards to bin count, we did not observe improved performance for 16384 over 8192 bins with respect to our quantitative metrics. In fact, we found that above these values, increasing bin counts may be detrimental to performance, as it leads to more unstable training and a larger memory footprint.
>
> > you first need to know the maximum extent of the molecule.
>
> Yes, we need to know the maximum extent of the molecule for discrete binning. We view this as a hyperparameter. Currently, generalization to larger systems is limited for all methods, despite baselines not employing binning strategies.
>
> ### On performance relative to flow models
> We agree that predicting each atom coordinate can be computationally demanding, but highlight that this is not much of a problem given the current LLM infrastructure, which has been scaled to work well with large context lengths. We also note that this approach compares well relative to the current state-of-the-art generative methods based on flows (FALCON).
> In Figure 1 / 16-A we would like to clarify that Robin clearly outperforms Prose on Energy-W2 and Torus-W2. TICA-W2 is less clear; this is due to the overall noise in this metric, which is apparent in Figures A.16-19 as new modes are discovered and correctly balanced with more samples.
>
> > In Table 2, the improvements over Prose only seem marginal.
>
> We wish to politely push back that the improvements over Prose in Table 2 are marginal. We believe Robin shows clear benefits over Prose, especially on larger systems with improvements in all metrics on the largest 8AA systems,  especially energy-W2, which achieves a 60% reduction over state-of-the-art. This result suggests that ArBG is more well-suited for larger systems and future scaling efforts.
>
> ### Interatomic distance plot
> We now plot interatomic distances following the procedure of previous work [1] in Fig. R1. We note that interatomic distances are relatively easy to fit. We find that the bin sizes we use are acceptable at this scale to model bond lengths well.
> ### Error accumulation over sequence length
> We acknowledge that error accumulation could be a problem. Our current models appear tolerant to error accumulation on current benchmarks up to the scale of 8-10AA. However, we do agree that this may become a problem on even larger systems, and therefore advocate for the investigation of other parameterizations as important directions for future work. Despite this, we highlight that this is not as much of an issue in autoregressive models as might be expected, as can be seen in the LLM setting. Finally, we also believe SMC-based methods, which we unlock through the AR parametrization of ArBG, allow for correcting errors upon detection, partially addressing this issue.
> ### Code release
> We commit to releasing full code, data, and experimental configurations upon acceptance.
> ### ALDP overfitting
> In Table R1 of the link, we include the training and validation loss for ArBG on alanine dipeptide, demonstrating model overfitting. It can be seen that around epoch 239, the training loss continues to decrease, while the validation loss begins to increase. We believe a variety of methods can be used to combat overfitting in this setting outside of early stopping. We did not use other regularization methods as we are primarily interested in larger systems.
>
> ### Concluding remarks
>
> We thank the reviewer again for their time and effort during this rebuttal process. We hope our answers here have aided in clarifying all the great questions raised by the reviewer. If there are any further questions held by the reviewer, we remain eager to engage further. If the reviewer feels satisfied with our responses and new rebuttal results, we would be thrilled if the reviewer would consider a fresher evaluation of our work with a potential score upgrade.
>
> ### References
>
> [1] Klein, Leon, and Frank Noé. "Transferable boltzmann generators." Advances in Neural Information Processing Systems 37 (2024): 45281-45314.

---

> > ### Author Rebuttal · Reviewer_6th3 · 2026-04-02
> >
> > I would like to thank the authors for their detailed response and thorough answers.
> >
> > I think that the authors have fully addressed all of my concerns, especially regarding performance and their binning strategy. I am grateful that the authors commit to releasing their source code, increasing my confidence in their work and reproducibility.
> >
> > I hope that the authors revise the parts in the text and add the additional/extended plots they provided. Especially, the additional analysis of alanine dipeptide overfitting.
> >
> > Overall, I increase my score to "Accept".

---

> > > ### Author Response · Authors · 2026-04-06
> > >
> > > We would like to thank the reviewer for their insightful comments and engagement in this discussion. We are glad that we could fully address all of the reviewer's concerns. We will include all additional experiments performed during this discussion in the updated version of the manuscript. We look forward to releasing the code and appreciate your detailed review, which has strengthened the rigor and increased the clarity of our work.

---

### Decision · Program_Chairs · 2026-04-30

**Decision:**

Accept (spotlight)

**Comment:**

This paper is concerned with the problem of sampling from a Boltzmann distribution. Many existing methods focus on flow-based methods. This paper proposes Autoregressive Boltzmann Generators, which is based on an autoregressive framework, and demonstrates its performance is better than flow-based approaches.

After the rebuttal discussion, reviewers unanimously have recommending accepting this paper. Thus, I am happy to recommend accepting the paper.